# Separating phases of allopolyploid evolution with resynthesized and natural *Capsella bursa-pastoris*

**Tianlin Duan[1]\*, Adrien Sicard[2], Sylvain Glémin[1,3], Martin Lascoux[1]\***

[1]Department of Ecology and Genetics, Evolutionary Biology Centre and Science for Life Laboratory, Uppsala University, Uppsala, Sweden; [2]Department of Plant Biology, Swedish University of Agricultural Sciences, Uppsala, Sweden; [3]UMR CNRS 6553 ECOBIO, Campus Beaulieu, Rennes, France

**\*For correspondence:**
tianlin.duan42@gmail.com (TD);
martin.lascoux@ebc.uu.se (ML)

**Competing interest:** The authors declare that no competing interests exist.

**Abstract** Allopolyploidization is a frequent evolutionary transition in plants that combines whole-genome duplication (WGD) and interspecific hybridization. The genome of an allopolyploid species results from initial interactions between parental genomes and long-term evolution. Distinguishing the contributions of these two phases is essential to understanding the evolutionary trajectory of allopolyploid species. Here, we compared phenotypic and transcriptomic changes in natural and resynthesized *Capsella* allotetraploids with their diploid parental species. We focused on phenotypic traits associated with the selfing syndrome and on transcription-level phenomena such as expression-level dominance (ELD), transgressive expression (TRE), and homoeolog expression bias (HEB). We found that selfing syndrome, high pollen, and seed quality in natural allotetraploids likely resulted from long-term evolution. Similarly, TRE and most down-regulated ELD were only found in natural allopolyploids. Natural allotetraploids also had more ELD toward the self-fertilizing parental species than resynthesized allotetraploids, mirroring the establishment of the selfing syndrome. However, short-term changes mattered, and 40% of the cases of ELD in natural allotetraploids were already observed in resynthesized allotetraploids. Resynthesized allotetraploids showed striking variation of HEB among chromosomes and individuals. Homoeologous synapsis was its primary source and may still be a source of genetic variation in natural allotetraploids. In conclusion, both short- and long-term mechanisms contributed to transcriptomic and phenotypic changes in natural allotetraploids. However, the initial gene expression changes were largely reshaped during long-term evolution leading to further morphological changes.

## eLife assessment

This **important** study offers new insight into how floral and reproductive phenotypes and gene expression evolve in allopolyploids. The authors marshal **compelling** evidence, using well-constructed genetic lines, RNA sequencing, and phenotypic analyses to distinguish the roles of hybridization, whole genome duplication, and subsequent evolution in phenotypes associated with the selfing syndrome and in gene expression. The work will be of interest to researchers working in plant speciation and genomics, as well as those more broadly interested in the effects of genome copy number on phenotypic and expression evolution.

## Introduction

Allopolyploidization is the coupling of whole genome duplication and interspecific hybridization, resulting in organisms possessing two or more diverged genomes. This intriguing evolutionary

transition is widespread in nature (**Albertin and Marullo, 2012**; **Barker et al., 2016**) and is of agricultural importance (**Behling et al., 2020**). Allopolyploidization is expected to have both short-term and long-term consequences: not only can the merging of divergent genomes itself be seen as a macro-mutation but also it triggers subsequent genomic changes over distinct time scales.

Right after allopolyploidization or within a few generations, various genomic and transcriptomic changes can be caused by a series of mechanisms, including DNA methylation repatterning (**Edger et al., 2017**; **Li et al., 2019**), reactivation of transposable elements (TE, reviewed in **Vicient and Casacuberta, 2017**), chromosome rearrangements, including homoeologous exchanges (**Parisod et al., 2009**; **Szadkowski et al., 2010**; **Lashermes et al., 2014**; **Xiong et al., 2021**) and intergenomic interactions between regulatory elements (**Shi et al., 2012**; **Hu and Wendel, 2019**). These multifaceted effects were initially proposed to be dramatic but are likely smoother and more subtle than initially thought. The fact remains that these short-term mechanisms add further complexity to the genetic variation gathered from parental lineages. Genetic changes can reinforce some initial epigenetic changes, leading to long-term heritable consequences in established allopolyploids. For instance, an epigenetically downregulated/silenced gene copy is more likely to degenerate than the other copy due to weaker purifying selection.

Apart from instant genomic changes, allopolyploidization also alters multiple genetic attributes, impacting the long-term evolution of allopolyploid genomes. First, as a minority cytotype, newly formed allopolyploid populations often experience a bottleneck (**Levin, 1975**; **Novikova et al., 2017**; **Griffiths et al., 2019**). This bottleneck reduces genetic variation within allopolyploid species and favors the fixation of neutral or slightly deleterious mutations (**Novikova et al., 2017**). Second, with an extra genome, allotetraploid species could undergo a period of relaxed purifying selection (**Lynch and Conery, 2000**; **Douglas et al., 2015**; **Paape et al., 2018**). Relaxed selection also accelerates the accumulation of deleterious mutations on allopolyploid genomes. At the same time, it facilitates neofunctionalization by allowing functional mutations to accumulate in one paralog while maintaining the ancestral function through the second (**Ohno, 1970**). Third, allopolyploidization immediately distorts both the relative and absolute dosage of gene product, which further alters physiological balance and efficiency (**Anneberg and Segraves, 2020**; **Yu et al., 2021**; **Domínguez-Delgado et al., 2021**). In the long term, both relative and absolute dosages of gene expression of allopolyploid genomes are expected to be under selection (**Bekaert et al., 2011**), and gradually adapt to a polyploid or hybrid state (**Bomblies, 2020**). Under the joint action of these forces, allopolyploid subgenomes are further co-adapted and degenerated, and subgenomes are often biasedly retained, termed biased fractionation (**Schnable et al., 2011**; **Tang et al., 2012**; **Renny-Byfield et al., 2015**; **Wendel et al., 2018**).

Both short-term reactions and long-term evolution can generate novel evolutionary opportunities and potentially allow allopolyploid lineages to have advantages in adaptation to novel environments (**Baniaga et al., 2020**). In established allopolyploids, phenomena caused by long-term evolutionary forces can be confounded by traces of short-term genomic changes. The relative contributions of short- and long-term mechanisms to genomic changes in allopolyploids can be assessed by comparing established natural allopolyploids with resynthesized allopolyploids (e.g. **Wang et al., 2006**; **Wang et al., 2016**; **Buggs et al., 2011**; **Yoo et al., 2013**; **Zhang et al., 2016b**).

Variation and novelties in gene expression caused by allopolyploidization are often assessed by homoeolog expression bias (HEB) and non-additive gene expression (**Grover et al., 2012**; **Yoo et al., 2013**; **Zhang et al., 2016a**; **Wu et al., 2018**; **Shan et al., 2020**). HEB measures the separate contributions of gene copies from different parental species (homoeologs) and non-additive gene expression measures the deviation of the total expression of both homoeologs from an intermediate value between parental species. Non-additive patterns of gene expressions are further classified as expression level dominance (ELD) and transgressive expression (TRE). ELD means that the total expression of both homoeologs is similar to the expression level of only one parental species (**Grover et al., 2012**), but differs from the expression level of the other. TRE means that gene expression in allopolyploids is higher or lower than in both parental species. Variation of HEB and non-additive gene expression in allopolyploids can be triggered by several mechanisms in the early generations of the new allopolyploid; or alternatively, they may arise during long-term evolution due to either neutral or selective processes.

*Capsella bursa-pastoris* is a natural allotetraploid plant species which originated about 100,000 years ago (**Douglas et al., 2015**). Two diploid species, *C. orientalis* and *C. grandiflora*, are extant relatives of

the maternal and paternal progenitors (hereinafter referred to as parental species) of *C. bursa-pastoris*, respectively (**Hurka et al., 2012**; **Douglas et al., 2015**). *C. grandiflora* is self-incompatible (SI), but *C. orientalis* was already self-compatible (SC) before the formation of *C. bursa-pastoris* (**Bachmann et al., 2019**). *C. bursa-pastoris* is also a self-compatible species, with typical selfing-syndrome characteristics. In particular, it has smaller petals, fewer pollen grains, and shorter styles than the outcrossing *C. grandiflora* (**Neuffer and Paetsch, 2013**). Yet, it remains unclear whether the inconspicuous flower phenotypes of *C. bursa-pastoris* only reflect the dominance relationship of the parental alleles or if these traits have also evolved post-allopolyploidization.

Natural *C. bursa-pastoris* exhibits disomic inheritance (**Hurka et al., 1989**; **Roux and Pannell, 2015**), with which chromosomes only recombine and segregate with their homologs during meiosis, but not with homoeologs. However, the strictness of disomic inheritance in *C. bursa-pastoris* has not been tested. In general, the two subgenomes of *C. bursa-pastoris* are still well-retained and functional. There is no sign of large-scale gene loss or silencing, although purifying selection has been weaker genome-wide (**Douglas et al., 2015**), and the *C. orientalis*-derived subgenome (Cbp_co) has accumulated more putatively deleterious mutations than the *C. grandiflora*-derived subgenome (Cbp_cg), both before and after the formation of *C. bursa-pastoris* (**Douglas et al., 2015**; **Kryvokhyzha et al., 2019a**). The majority of genes are expressed from both homoeologs, and on average there is only a slight HEB toward Cbp_cg homoeologs (**Douglas et al., 2015**; **Kryvokhyzha et al., 2019a**). Most genes are additively expressed in natural *C. bursa-pastoris*, but ELD and transgressive gene expression have also been observed (**Kryvokhyzha et al., 2019a**). Despite the moderate HEB and non-additive expression, gene expression in *C. bursa-pastoris* showed some striking tissue-specific features (**Kryvokhyzha et al., 2019a**). In flowers, gene expression levels in *C. bursa-pastoris* resembled those in *C. orientalis*, while in leaves and roots, gene expression levels were more similar to those in *C. grandiflora*.

In contrast to the drastic genomic or transcriptomic changes observed in allopolyploid wheat (**Zhang et al., 2016a**), *Brassica* (**Szadkowski et al., 2010**; **Lloyd et al., 2018**), and *Tragopogon* (**Chester et al., 2012**), natural *C. bursa-pastoris* represents another paradigm where established allopolyploid species show only mild genomic changes and expression bias. This contrast raises questions. Was the genome of natural *C. bursa-pastoris* less affected by putative short-term mechanisms, or was it the result of 100,000 years' evolution, which filtered out or compensated for the initial drastic changes? What are the relative strengths of short-term mechanisms and long-term evolution in shaping genomic and phenotypic variation in allopolyploids?

Resynthesized allopolyploids are the closest approximation to the early stage of natural allopolyploids. They provide a reference point for separating the short-term effects of allopolyploidization from long-term evolutionary changes. The present study builds upon **Duan et al., 2023**, which showed that hybridization played a much larger role than whole genome doubling during the creation of resynthesized polyploids in the *Capsella* genus. Here, we compared transcriptomes and phenotypes of resynthesized *C. bursa-pastoris*-like allotetraploids with natural *C. bursa-pastoris* and its two diploid progenitors. We focused on teasing apart the contributions from short- and long-term processes to (1) phenotypes, (2) non-additive gene expression, and (3) HEB in *Capsella* allotetraploids.

## Results

### The selfing syndrome was observed in natural C. bursa-pastoris but not in resynthesized allotetraploids

The breakdown of self-incompatibility in allotetraploid *Capsella* can directly result from hybridizing with the self-fertilizing species (**Bachmann et al., 2021**; **Duan et al., 2023**). We explored to what extent the development of a selfing syndrome was instantly achieved after allopolyploidization or, instead, developed later on by comparing phenotypes of resynthesized allotetraploids (groups Sd and Sh), natural *C. bursa-pastoris* (Cbp) and the diploid parental species, *C. grandiflora* (Cg2) and *C. orientalis* (Co2). The resynthesized allotetraploids were generated with individuals from one population of each diploid parental species (*Figure 1*), and the Sd ('WGD-first') and Sh ('hybridization-first') groups only differed in the order of WGD and hybridization (**Duan et al., 2023**). There were six 'lines' in each of the five plant groups. For the Sd and Sh groups, each line represented an independent allopolyploidization event, while the six lines of natural *C. bursa-pastoris* were from six different

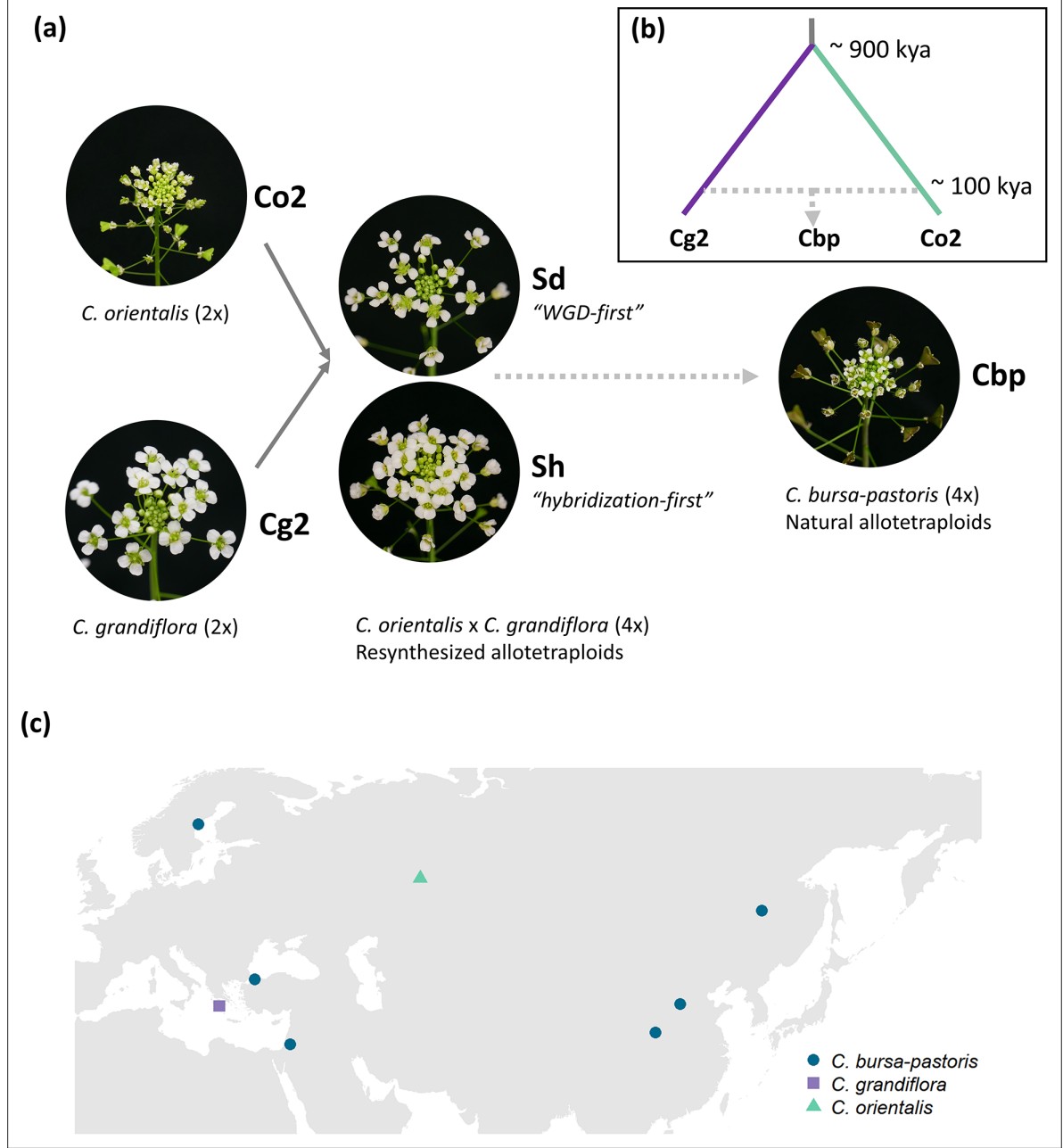

**Figure 1.** Plant material used in the present study. (**a**) Five groups of *Capsella* plants. Diploid species (Co2 and Cg2 groups) and the second generation of resynthesized allotetraploids (Sd and Sh groups) were from *Duan et al., 2023*. Samples of natural allotetraploids, *C. bursa-pastoris*, were added to the present study. (**b**) Phylogenetic relationship of the three natural species used in the present study, modified from *Douglas et al., 2015*; *C. bursa-pastoris* originated from the hybridization between the ancestral population of *C. orientalis* and the (*C. grandiflora* + *C. rubella*) lineage, and *C. rubella* were omitted from the figure; kya: thousand years ago. (**c**) Geographic origin of the *Capsella* samples.

The online version of this article includes the following source data for figure 1:

**Source data 1.** *Capsella* plants used in the present study.

populations (*Figure 1c* and *Figure 1—source data 1*), representing three major genetic clusters of the wild *C. bursa-pastoris* (*Kryvokhyzha et al., 2016*). For diploid parental groups, a line referred to a full-sibling family resulting from self-fertilization (Co2) or one controlled cross (Cg2). Phenotypes of the five groups were measured in a growth chamber on about 36 individuals per plant group (6 individuals × 6 lines).

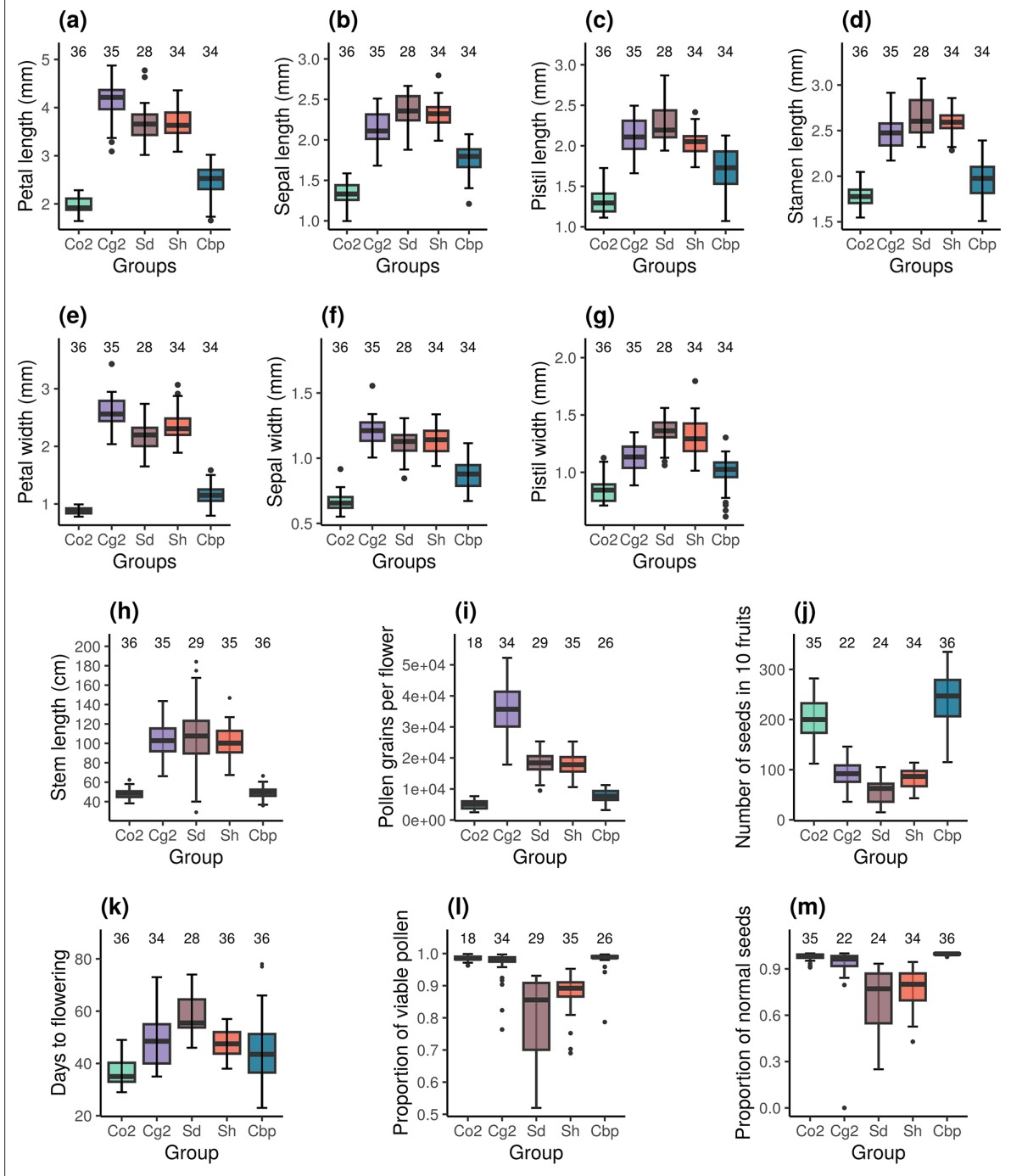

**Figure 2.** Phenotypic traits of the five *Capsella* groups. Co2: diploid *C. orientalis*; Cg2: diploid *C. grandiflora*; Sd: whole-genome-duplication-first resynthesized allotetraploids; Sh: hybridization-first resynthesized allotetraploids; Cbp: natural allotetraploid *C. bursa-pastoris*. The measured traits were (**a**) petal length, (**b**) sepal length, (**c**) pistil length, (**d**) stamen length, (**e**) petal width, (**f**) sepal length, (**g**) pistil width, (**h**) length of the longest stem, (**i**) number of pollen grains per flower, (**j**) number of seeds in ten fruits, (**k**) number of days from germination to the opening of the first flower, (**l**) proportion of viable pollen grains, and (**m**) proportion of normal seeds in ten fruits. Sample sizes are shown above the groups.

The online version of this article includes the following source data for figure 2:

**Source data 1.** Effects of plant group and positions (tray ID) on phenotypes.

Resynthesized and natural allotetraploids had distinct floral morphologies (*Figure 2a–g*). Indeed, natural allotetraploids had significantly shorter and narrower petals, sepals and pistils and shorter stamina than resynthesized allotetraploids (one-way ANOVA, $F_{4,160} > 78$ and $p<0.001$ in all seven tests; Tukey's HSD test, $\alpha=0.01$). Pollen and seed production was also affected. Natural allotetraploids had fewer pollen grains per flower (*Figure 2i*). While the number of pollen grains in resynthesized allotetraploids was intermediate between the two parental species, the number of pollen grains of the natural allotetraploid group was now similar to that of the Co2 group (one-way ANOVA, $F_{4,137} = 164.6$, $p<0.001$; Tukey's HSD test, $\alpha=0.01$). Moreover, the number of seeds per fruit in natural allotetraploids was much larger than in resynthesized allotetraploids. The resynthesized allotetraploid groups had a similar number of seeds in 10 fruits to that of the Cg2 group, whereas the number of seeds in 10 fruits in the natural allotetraploid group was even higher than that of the Co2 group (*Figure 2j*; one-way ANOVA, $F_{4,146} = 152.5$, $p<0.001$; Tukey's HSD test, $\alpha=0.01$).

The architecture and phenology of the whole plant were affected too. The stem length of natural Cbp was shorter than in resynthesized allotetraploids but was similar to the stem length of the Co2 group (*Figure 2h*; one-way ANOVA, $F_{4,166} = 84.5$, $p<0.001$; Tukey's HSD test, $\alpha=0.01$). Finally, plants of the Cbp group flowered earlier than those of the Sd group, but at a similar time as those of the Sh group (*Figure 2k*; one-way ANOVA with hc3 White's correction, $F_{4,165} = 49.2$, $p<0.001$; Tukey's HSD test, $\alpha=0.01$).

## Pollen viability and seed quality improved in natural *Capsella* allotetraploids

Pollen viability and the proportion of normal seeds were compared between resynthesized and natural allotetraploids. For both traits, we observed a decrease in pollen viability in resynthesized allotetraploids followed by recovery in natural Cbp. Both Sd and Sh groups had lower proportions of viable pollen than the diploid parental species, but the proportion of viable pollen in natural Cbp was similar to that of the diploid parental species (*Figure 2l*; GLM, quasi binomial, $F_{4,137} = 24.4$, $p<0.001$; Tukey's HSD test, $\alpha=0.01$). The resynthesized allotetraploids generated a higher proportion of abnormal seeds than the three natural species (*Figure 2m*; GLM, quasi binomial, $F_{4,146} = 59.2$, $p<0.001$; Tukey's HSD test, $\alpha=0.01$). The average percentage of normal seeds was 69.6±4.3% in the Sd group and 77.5±2.2% in the Sh group. In contrast, the natural Cbp had almost no abnormal seeds, with a percentage of normal seeds of 99.6±0.8%.

## A two-step evolution of the global expression pattern of natural allopolyploid Cbp

To compare the gene expression pattern of the five plant groups, RNA-sequencing was conducted for one individual per line and six lines per group, using young inflorescences (flowers) and leaves. Expression levels were determined for 21,937 genes in flower samples and 18,999 genes in leaf samples after excluding genes with CPM >1 in less than two samples. The overall gene expression pattern was visualized with multi-dimensional scaling (MDS) analysis (*Figure 3a and b*). For unphased gene expression, the resynthesized allotetraploids lay between the diploid parental species in both flowers and leaves. Natural Cbp samples were also intermediate between parental species in the first dimension but were far from the resynthesized allotetraploids in the other dimension, showing the effect of long-term evolution in Cbp and possibly also the divergence between extant diploid species and the real progenitors.

The expression levels of separate homoeologs in allotetraploids were determined with the diagnostic SNPs between the two diploid species. For the Sd, Sh, and Cbp groups, 52.8%, 53.7%, and 44.7% of the mapped reads could be assigned to one of the homoeologs, respectively. The expression pattern of each allotetraploid subgenome was more similar to the corresponding diploid progenitor (*Figure 3c and d*). The pattern of expression of resynthesized allotetraploids was intermediate between those of diploid progenitors and natural Cbp.

Differential expression analysis was performed among the five plant groups, using the downsampled unphased gene expression data. With a threshold of FC >2 and FDR <0.05, no significant differentially expressed genes (DEGs) were found between the two resynthesized allotetraploid groups, while 311–2888 DEGs were revealed in other group contrasts (*Figure 3—figure supplements 1 and 2*). There are two salient features. First, compared to either diploid progenitor, most DEGs in

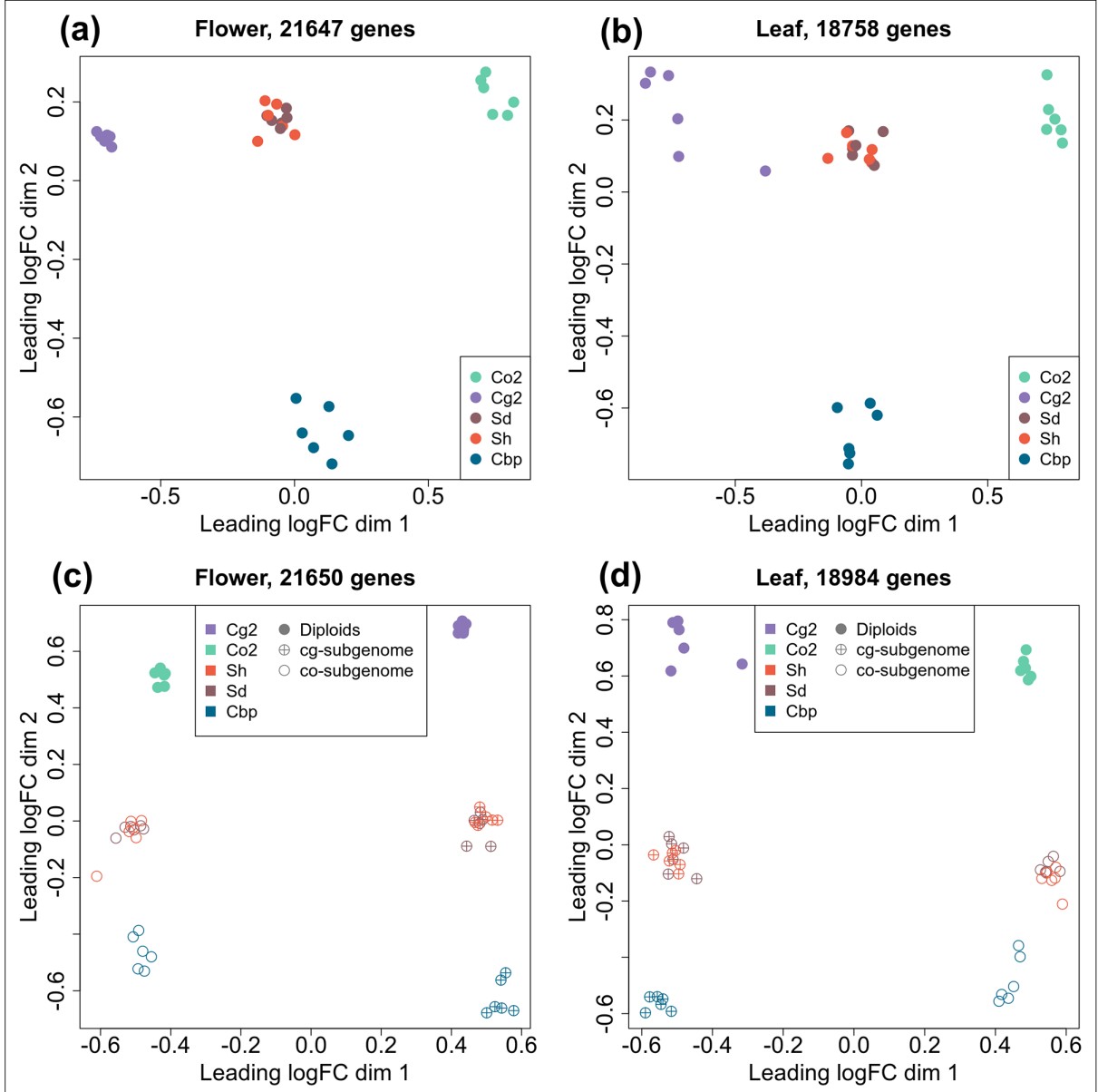

**Figure 3.** Multi-dimensional scaling (MDS) analyses of gene or homoeolog expression in two tissues. Down-sampled gene expressions were used to compare gene expression patterns of the five plant groups in either flowers (**a**) or leaves (**b**). The five groups of *Capsella* plants are: diploid parental species *C. grandiflora* (Cg2) and *C. orientalis* (Co2), hybridization-first (Sh) and whole-genome-duplication-first (Sd) resynthesized allotetraploids, and natural allotetraploid *C. bursa-pastoris* (Cbp). Separated homoeolog expressions in allotetraploids were then compared with rescaled gene expression of diploid groups in both flowers (**c**) and leaves (**d**). All the MDS analyses used genes with count-per-million (CPM) >1 in at least two samples, and expression levels were normalized with the trimmed mean of M-values (TMM) method.

The online version of this article includes the following figure supplement(s) for figure 3:

**Figure supplement 1.** Differentially expressed genes (DEGs) in pair-wise contrasts among the five *Capsella* plant groups in flowers and leaves.

**Figure supplement 2.** Summary of differential expression analyses of allotetraploid groups.

both resynthesized and natural allotetraploid groups were up-regulated (*Figure 3—figure supplement 2*). The proportion of down-regulated DEGs increased, nevertheless, in natural allotetraploids. Second, both resynthesized allotetraploids, Sd and Sh, have much more DEG with Co2 than with Cg2. However, this is no longer the case in Cbp where the two comparisons yielded similar results.

Although we could not make a clear expectation for gene ontology (GO) terms that would be overrepresented in DEGs between resynthesized and natural allopolyploids, we are not the only study

that compared newly resynthesized and established allopolyploids, and GO terms that were repeatedly revealed by exploratory analysis may give a hint for future studies. For this reason, we further performed a GO enrichment analysis, using genes that were differentially expressed in both Cbp-Sd and Cbp-Sh contrasts as the test set, and all expressed genes with GO annotations as the background set. In flowers or leaves, the top 10 most-enriched GO terms for biological processes were related to proteolysis, xenobiotic transport, regulation of translational fidelity, telomere maintenance and organization, DNA geometric change and duplex unwinding, cellular response to toxic substance, aminoacylation, peptidyl-lysine methylation, and heterochromatin formation and organization (*Supplementary file 3*). But after adjusting p-values with the Benjamini-Hochberg procedure, only GO term proteolysis (GO:0006508) was significantly overrepresented in DEGs between resynthesized and natural allotetraploids in leaves (Fisher's exact test, adjusted p-value = 0.0175).

## Both short- and long-term mechanisms contributed to expression level dominance in natural Cbp, but transgressive expressions were mainly from long-term evolution

Non-additive gene expression shared by natural allotetraploids may be triggered right after allopolyploidization by short-term deterministic mechanisms, such as intergenomic interactions of regulatory elements. Alternatively, non-additive expression may have been caused by mechanisms with stochastic effects or arose later during long-term evolution. We explored to what extent the non-additive expression shared by natural allotetraploids could reflect short-term deterministic mechanisms. By comparing gene expression levels in allotetraploids and diploid species, 21,647 genes in flowers and 18,758 genes in leaves were classified into one of the 10 expression categories, using the results of DE analysis on unphased gene expression (FC >2 and FDR <0.05). We focused on complete ELD and TREs: complete ELD is obtained when the gene expression level in an allopolyploid group is similar to that in one diploid group but not to the expression in the other diploid group, and TRE is detected when the gene expression level in an allopolyploid group is either higher or lower than in both diploid groups.

The percentage of genes showing complete ELD was altogether limited but doubled in natural allotetraploids relative to resynthesized allotetraploids (5.5% of genes in resynthesized allotetraploids and 10.2% in natural allotetraploids. *Figure 4a and b* and *Figure 4—source data 1*). Genes with ELD and the directions of ELD were highly shared between the two resynthesized allotetraploid groups (*Figure 4c*, *Figure 4—figure supplement 1*). The majority of these shared ELD were retained in natural allotetraploids (63.3% in flowers and 72.2% in leaves), suggesting that short-term deterministic mechanism contributed to ELD in natural allotetraploids. However, Cbp-specific ELD was also abundant, comprising more than half of the cases found in natural allotetraploids (56.6% in flowers and 60.8% in leaves), thereby showing the effects of long-term evolution.

The direction of ELD shifted between the resynthesized and natural allotetraploids (*Figure 4a and b* and *Figure 4—source data 1*). In resynthesized allotetraploids, most cases of ELD were up-regulated, and the number of genes with ELD toward *C. grandiflora* (Cg-ELD) was about twice of that toward *C. orientalis* (Co-ELD). Natural allotetraploids still had more up-regulated ELD than down-regulated ones, but the proportion of down-regulated cases increased. The proportion of Co-ELD had also increased in natural allotetraploids. In flowers, natural allotetraploids had more Co-ELD (1101) than Cg-ELD (938), and the number of Cg- and Co-ELDs were similar in leaves (Cg-ELD: 854, Co-ELD: 839).

Almost no TRE was found in resynthesized allotetraploids (less than five genes in either Sd or Sh group and in either tissue, *Figure 4a and b*, and *Figure 4—source data 1*). In contrast, about 1.3% of genes in Cbp showed TRE in both flowers and leaves.

## Segregation and recombination of homoeologous chromosomes were a major source of homoeolog expression bias variation in resynthesized *Capsella* allotetraploids

HEB of genes in an allotetraploid individual was measured by the ratio of expression of the *C. grandiflora*-origin homoeolog (cg) to the total expression of both homoeologs (cg/(cg +co)). To obtain a reliable gene expression ratio, lowly expressed genes (CPM(cg +co)<1 in any allotetraploid

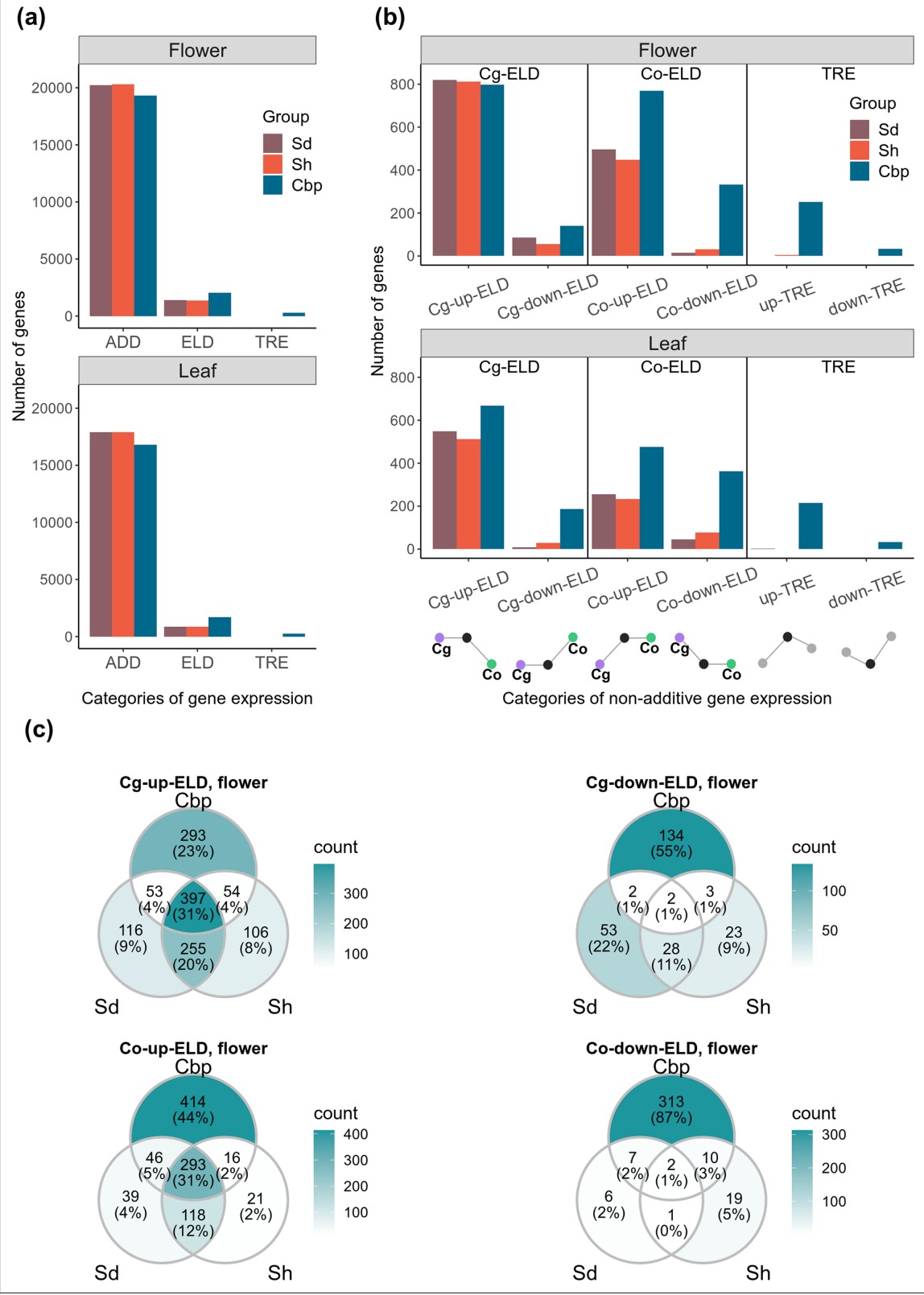

**Figure 4.** Additive and non-additive expression in allotetraploid groups. Sd: whole-genome-duplication-first resynthesized allotetraploids; Sh: hybridization-first resynthesized allotetraploids; Cbp: natural allotetraploid *C. bursa-pastoris*. (**a**) Number of genes that showed additive expression (ADD, including partial expression level dominance [ELD]), complete ELD, and transgressive expression (TRE) in each allotetraploid group. (**b**) Genes with complete ELD or TRE were further classified by whether they were up- or down-regulated in allotetraploids, and whether the expression level in

*Figure 4 continued on next page*

*Figure 4 continued*

allotetraploids was similar to *C. grandiflora* (Cg-ELD) or *C. orientalis* (Co-ELD). (**c**) Venn diagram of genes with complete ELD of the three allotetraploid groups in flowers, separated by directions of ELD.

The online version of this article includes the following source data and figure supplement(s) for figure 4:

**Source data 1.** Additive and non-additive gene expression in allotetraploid groups.

**Figure supplement 1.** Genes showed expression level dominance (ELD) of the three allotetraploid groups in leaves.

individual) were excluded from this analysis. Eventually, the homoeolog expression ratio was calculated for 18,255 genes in flowers, and 15,581 genes in leaves.

Overall, none of the three allotetraploid groups showed a strong average HEB. In flowers, the average HEB was 0.499±0.001, 0.531±0.001, and 0.475±0.001 for the Sd, Sh, and Cbp groups, respectively. In leaves, the average HEB was 0.504±0.001, 0.532±0.001, and 0.477 for the Sd, Sh, and Cbp groups. When averaged among individuals, the HEB of resynthesized allotetraploids had smaller gene-wise variation than that of the Cbp group (Levene's test, p-value <0.001).

Among resynthesized allotetraploids, although the average homoeolog expression ratio was not systematically biased toward Cg or Co, HEB showed great variation among chromosomes and individuals (***Figure 5a and b***, ***Figure 5—figure supplements 1 and 2***). The distribution of HEB in some chromosomes had peaks around 0, 0.25, 0.75, or 1, but the shape of the distribution was almost identical between flower and leaf samples. When HEB of genes was plotted along chromosomal positions, we found that the extra peaks in the distribution of HEB can be further explained by large genomic segments separated by a sudden change of average HEB (***Figure 5c***, ***Figure 5—figure supplements 3–6***). Altogether, the pattern suggested that some chromosomes or chromosomal regions in resynthesized allotetraploids had an unbalanced number of cg- and co-homoeologs (not 2:2), which were likely caused by the segregation and recombination of homoeologous chromosomes. Both the segregation and recombination of homoeologous chromosomes are outcomes of homoeologous synapsis (synapsis between homoeologous chromosomes during meiosis), which reflects polysomic or mixed inheritance in resynthesized allopolyploids. For short, we refer to both segregation and recombination of homoeologous chromosomes as homoeologous synapsis.

The effect of possessing an unbalanced number of homoeologs largely increased the variation of HEB in resynthesized allotetraploids. The breakpoint between segments with distinct average HEB and the copy number of cg-homoeolog on each segment were estimated with a five-state Hidden Markov Model (HMM), using HEB along chromosomes (***Figure 5c***, ***Figure 5—figure supplement 7***). Among the 96 chromosome quartets (two pairs of homologous chromosomes) from the 12 resynthesized allotetraploid individuals, only 39 chromosome quartets showed no sign of homoeologous synapsis, that is no breakpoint was identified and the estimated number of cg-homoeolog across the chromosome was two. On average 0.833±0.097 (mean ± se) breakpoint was identified for each chromosome quartet, and 31.0% of genes were estimated to have different numbers of cg- and co-homoeologs. Finally, for resynthesized allotetraploids, the estimated copy number of homoeologs was able to explain 48.4% and 46.8% of the variance of HEB in flowers and leaves, respectively (GLM with quasi-binomial error distribution, p<0.001 in both tissues).

In contrast to resynthesized allotetraploids, the distribution of HEB of natural Cbp was similar among individuals and chromosomes (***Figure 5a and b***, ***Figure 5—figure supplement 2***), although the distribution of HEB of some chromosomes of Cbp also showed weak bumps around 0, 0.25, 0.75, or 1 (***Figure 5b***, ***Figure 5—figure supplement 2***). We could not confidently estimate the number of homoeologs or the breakpoint of segments for Cbp with only RNA-sequencing data, as segments resulting from homoeologous exchanges could be shorter in natural Cbp, and the signals of copy number could be blurred by the variance of regulatory divergence. Nevertheless, the HMM segmentation algorithm also identified some candidate segments of which the average HEB strongly deviated from 0.5. Some candidate segments were only shared by individuals from the same population (***Figure 6***, ***Figure 6—figure supplements 1 and 2***).

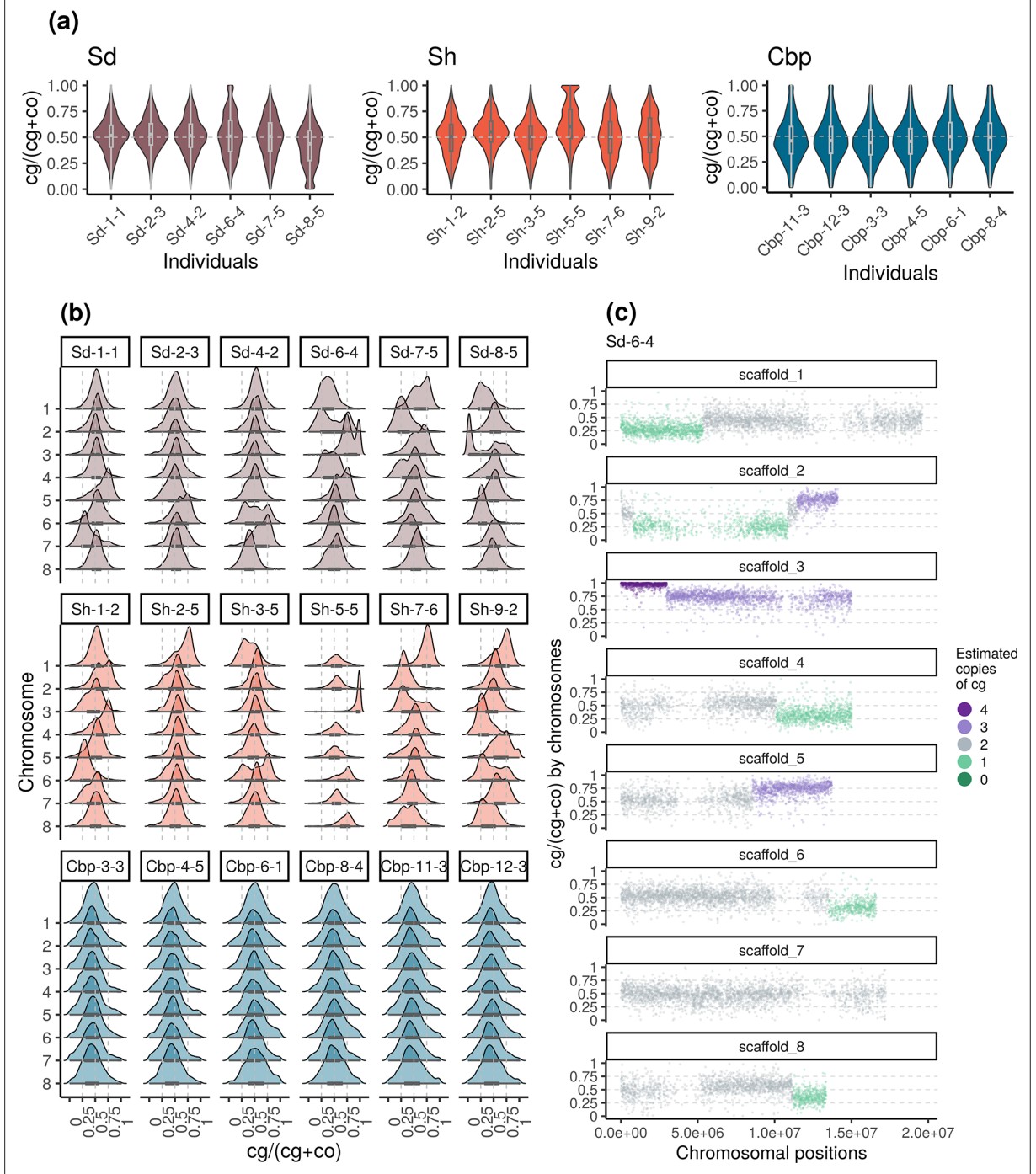

**Figure 5.** Variation of homoeolog expression bias (HEB) of the three allotetraploid groups in flowers. Sd: whole-genome-duplication-first resynthesized allotetraploids; Sh: hybridization-first resynthesized allotetraploids; Cbp: natural allotetraploid *C. bursa-pastoris*. Gene-wise HEB was calculated as the expression level of cg-homoeolog divided by the total expression level of both cg- and co- homoeologs (cg/(cg +co)). For each individual, HEB was calculated for 18,255 genes, which had count-per-million>1 in all flower samples. The distribution of HEB was shown by (**a**) individuals and (**b**) chromosomes. (**c**) HEB was also plotted along chromosome positions to show the sudden change of average HEB between genomic blocks, taking individual Sd-6–4 as an example. The number of cg-homoeologs at each gene estimated by the five-state Hidden Markov Model (HMM) was indicated by five colors. Dark green, light green, grey, light purple, and dark purple represent (0, 1, 2, 3, 4) cg-homoeologs and (4, 3, 2, 1, 0) co-homoeologs, respectively.

The online version of this article includes the following figure supplement(s) for figure 5:

**Figure supplement 1.** Distribution of gene homoeolog expression bias (HEB) by individuals.

*Figure 5 continued on next page*

*Figure 5 continued*

**Figure supplement 2.** Distribution of gene homoeolog expression bias (HEB) by chromosomes in leaves.

**Figure supplement 3.** Homoeolog expression bias along chromosome positions in the inflorescence sample of the Sd group ('Whole-genome-duplication-first'" resynthesized *Capsella* allotetraploids).

**Figure supplement 4.** Homoeolog expression bias along chromosome positions in the leaf sample of the Sd group ('Whole-genome-duplication-first' resynthesized *Capsella* allotetraploids).

**Figure supplement 5.** Homoeolog expression bias along chromosome positions in the inflorescence sample of the Sh group ('hybridization-first' resynthesized *Capsella* allotetraploids).

**Figure supplement 6.** Homoeolog expression bias along chromosome positions in the leaf sample of the Sh group ('hybridization-first' resynthesized *Capsella* allotetraploids).

**Figure supplement 7.** Estimated number of breakpoints per chromosome quartet in resynthesized allotetraploids.

## Most resynthesized allotetraploids had less homoeolog expression loss than natural Cbp, but with extreme outliers

Loss of homoeolog expression is a common phenomenon in allopolyploid species, which can be caused by homoeolog loss or silencing (*Buggs et al., 2009*; *Cox et al., 2014*; *Lashermes et al., 2016*). Loss of homoeolog expression may quickly arise after allopolyploidy, or alternatively, reflect a gradual biased gene fractionation during diploidization. We compared the extent of loss of homoeolog expression between resynthesized and natural *Capsella* allotetraploids, and among allotetraploid individuals.

The loss of homoeolog expression was identified from genes with medium to high expression levels in all individuals of the corresponding diploid species (*Figure 7*) to reduce noise from RNA-sequencing and phasing. On average, only 1.0% of these genes showed homoeolog-specific expression loss in natural Cbp. Most resynthesized allotetraploids have a lower level of homoeolog-specific expression loss than natural Cbp, but three individuals (Sd-6–4, Sd-8–5, and Sh-5–5) showed an extremely high level of homoeolog expression loss. The striking homoeolog expression loss in these three resynthesized allotetraploids was most likely caused by the segregation and recombination of homoeologous chromosomes, as the extreme HEB in the three outliers was restricted to chromosome 3, where the entire chromosome or a large chunk of the chromosome has only expression from one homoeolog (*Figure 5b*, *Figure 5—figure supplements 3–6*).

## Expression level dominance is caused by different mechanisms in resynthesized and natural allotetraploids

As homoeologous synapsis seemed to be a major cause of HEB and homoeolog-specific expression loss in resynthesized allotetraploids, we assessed whether it could have also played a role in the evolution of ELD. To do so, we explored the mechanism of ELD in resynthesized allotetraploids by comparing the gene expression change of separate homoeologs relative to the corresponding gene in diploid groups (log2FC(cg/Cg2) and log2FC(co/Co2)) among non-additive gene expression categories (*Figure 8*).

For ELD in resynthesized allotetraploids, different non-exclusive short-term mechanisms would produce different patterns of average expression change of EL-dominant (homoeolog derived from the diploid progenitor to which the total expression of both homoeologs was similar) and EL-recessive homoeologs (the opposite homoeolog), among genes with significant ELD: (i) If ELD was mainly caused by possessing more than two copies of the EL-dominant homoeolog (due to the segregation or recombination of homoeologous chromosomes), we would expect on average the expression of EL-dominant homoeolog to increase, and EL-recessive homoeolog to decrease, in both up- and down-regulated cases of ELD. (ii) If ELD was mainly caused by mechanisms with random effects, such as TE transposition, on average, there should be no large difference between the expression changes of EL-dominant and EL-recessive homoeologs. Because the occurrence and regulatory effects of new TE transpositions do not depend on the original relative expression level of the two homoeologs, that is the highly and lowly expressed homoeologs are equally likely to be up- or down-regulated by new TE transpositions. (iii) Predictions for new intergenomic interactions of regulatory elements can be complex, but under a simple scenario (*Hu and Wendel, 2019*), ELD may be caused by divergent trans-acting factors. In allopolyploids, stronger trans-acting factors act on the cis-regulatory elements of the opposite homoeolog, causing a similar regulatory effect if transcription rate were not limited by

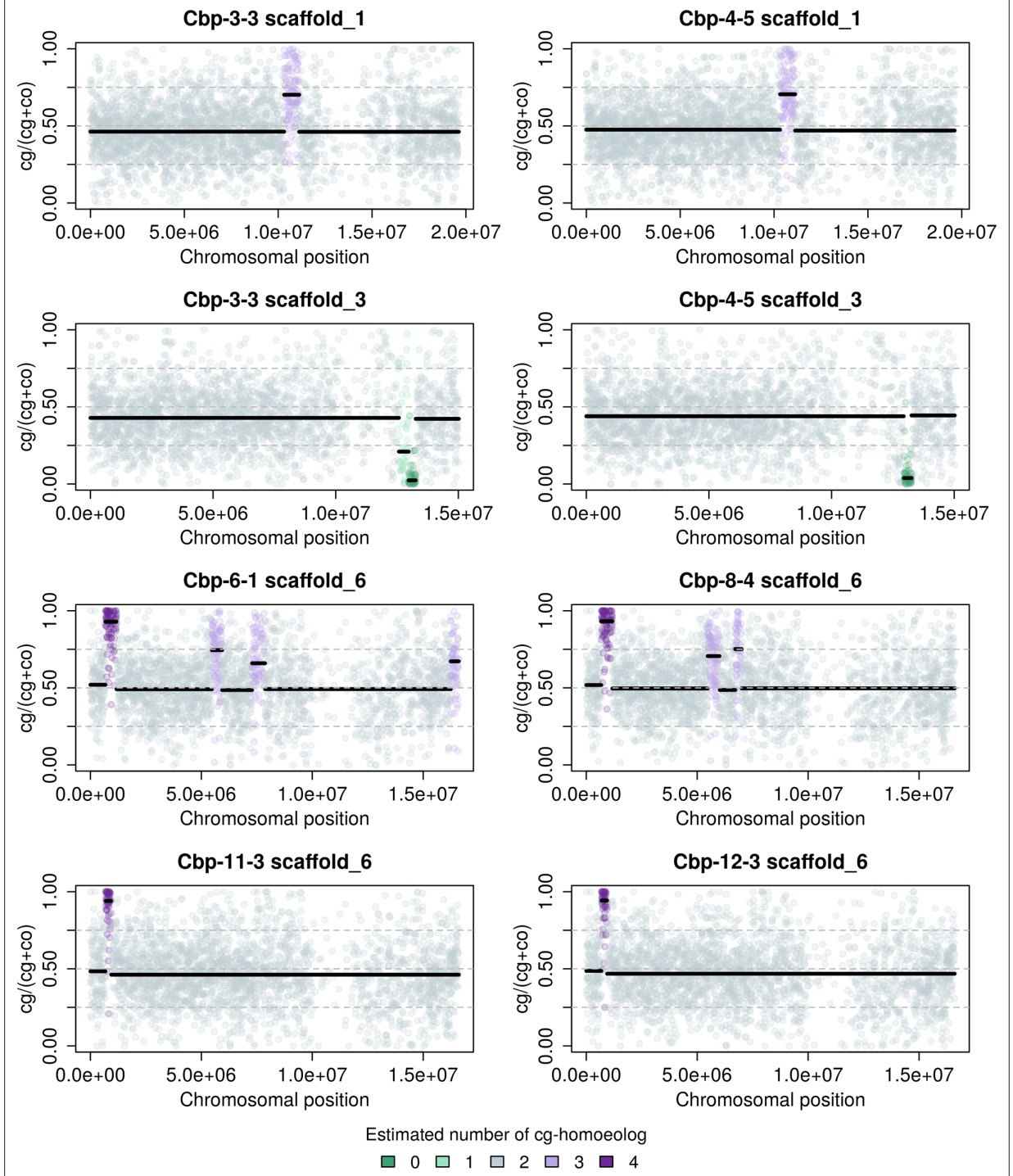

**Figure 6.** Homoeolog expression bias (cg/(cg +co)) along chromosomes of natural allotetraploid *C. bursa-pastoris* in flowers, taking four pairs of chromosome quartets with typical patterns as an example. The number of cg-homoeologs estimated by the five-state Hidden Markov Model was indicated by five colors. The two chromosome quartets in the same row are from the two individuals of the same major genetic cluster of natural *C. bursa-pastoris* (**Kryvokhyzha et al., 2016**), showing that some estimated segments with an unbalanced number of cg- and co-homoeologs were shared between the individuals from the same genetic cluster.

The online version of this article includes the following figure supplement(s) for figure 6:

**Figure supplement 1.** Homoeolog expression bias along chromosome positions in the inflorescence sample of the Cbp group (natural *Capsella bursa-pastoris*).

**Figure supplement 2.** Homoeolog expression bias along chromosome positions in the leaf samples of the Cbp group (natural *Capsella bursa-pastoris*).

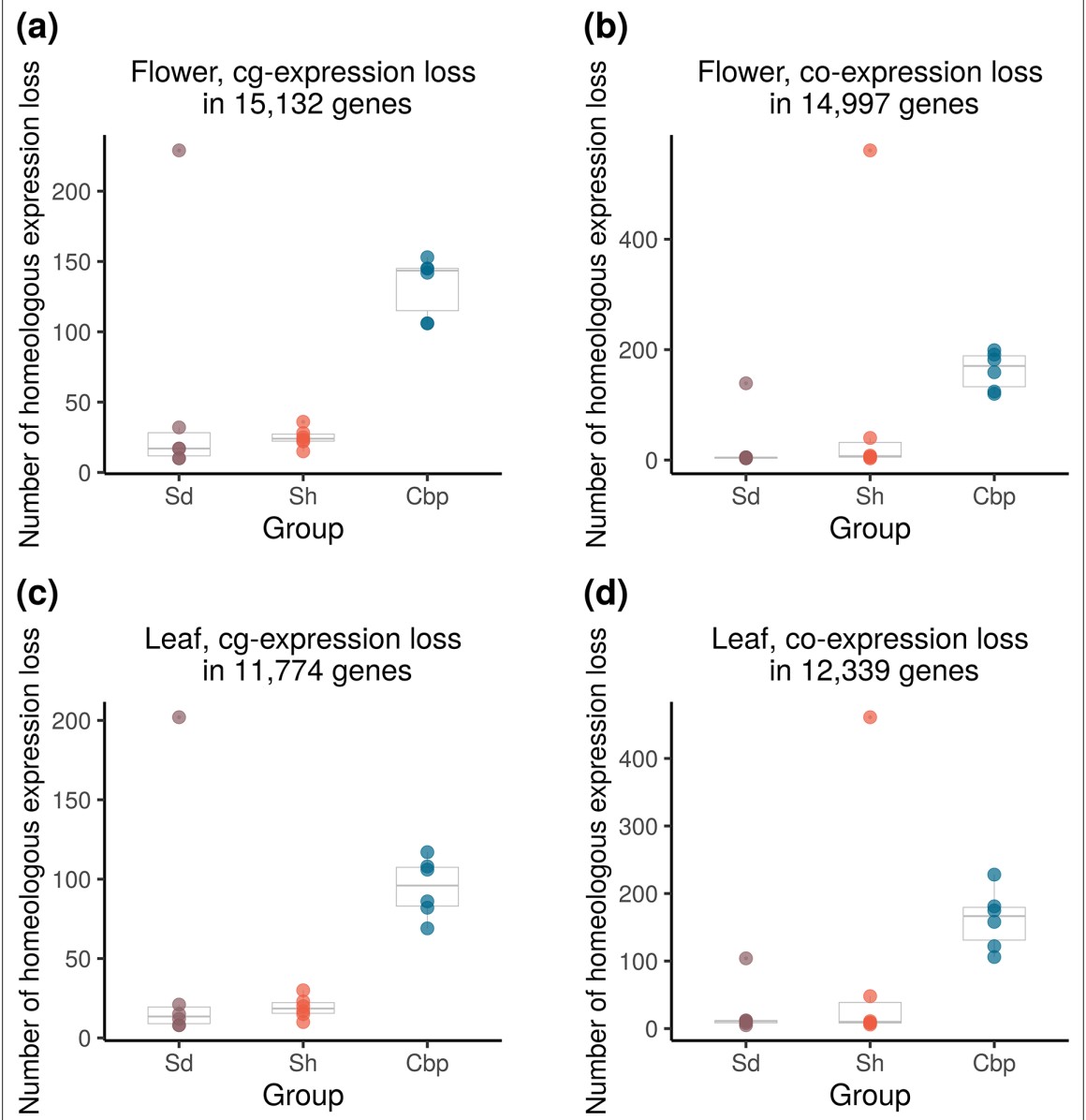

**Figure 7.** Loss of homoeolog expression in resynthesized (Sd and Sh) and natural allotetraploids (Cbp). The number of genes with the expression loss of *C. orientalis*-homoeolog (**a,c**) or *C. grandiflora*-homoeolog (**b,d**) per individual was compared among the three groups of allotetraploids in flowers (**a,b**) or leaves (**c,d**). Homoeologous genes that had obvious expression (count per million >5) in all individuals of the corresponding diploid species but almost no expression (count per million <0.5) in one allotetraploid individual were considered cases of homoeolog expression loss.

the concentration of these trans-regulators. If this mechanism were the main cause of ELD, on average the EL-recessive homoeolog should have the larger expression change in all categories of ELD, while the expression of EL-dominant homoeolog is not expected to change. In all other cases, we would not have direct inference on the exact mechanism of ELD, but at least the three mechanisms listed above could not be the predominant cause of ELD.

Concerning the ELD found in resynthesized allotetraploids, the change of homoeolog expression fits the third scenario. In all four categories of ELD, the EL-recessive homoeolog had a larger average expression change in the same direction as ELD, while the average expression change of EL-dominant homoeolog was closer to zero (*Figure 8*, *Figure 8—figure supplement 1*, and *Figure 8—source data 1*).

The Cbp-specific ELD showed a different pattern. Although the EL-recessive homoeolog still had a larger expression change, the EL-dominant also showed non-zero average expression change toward

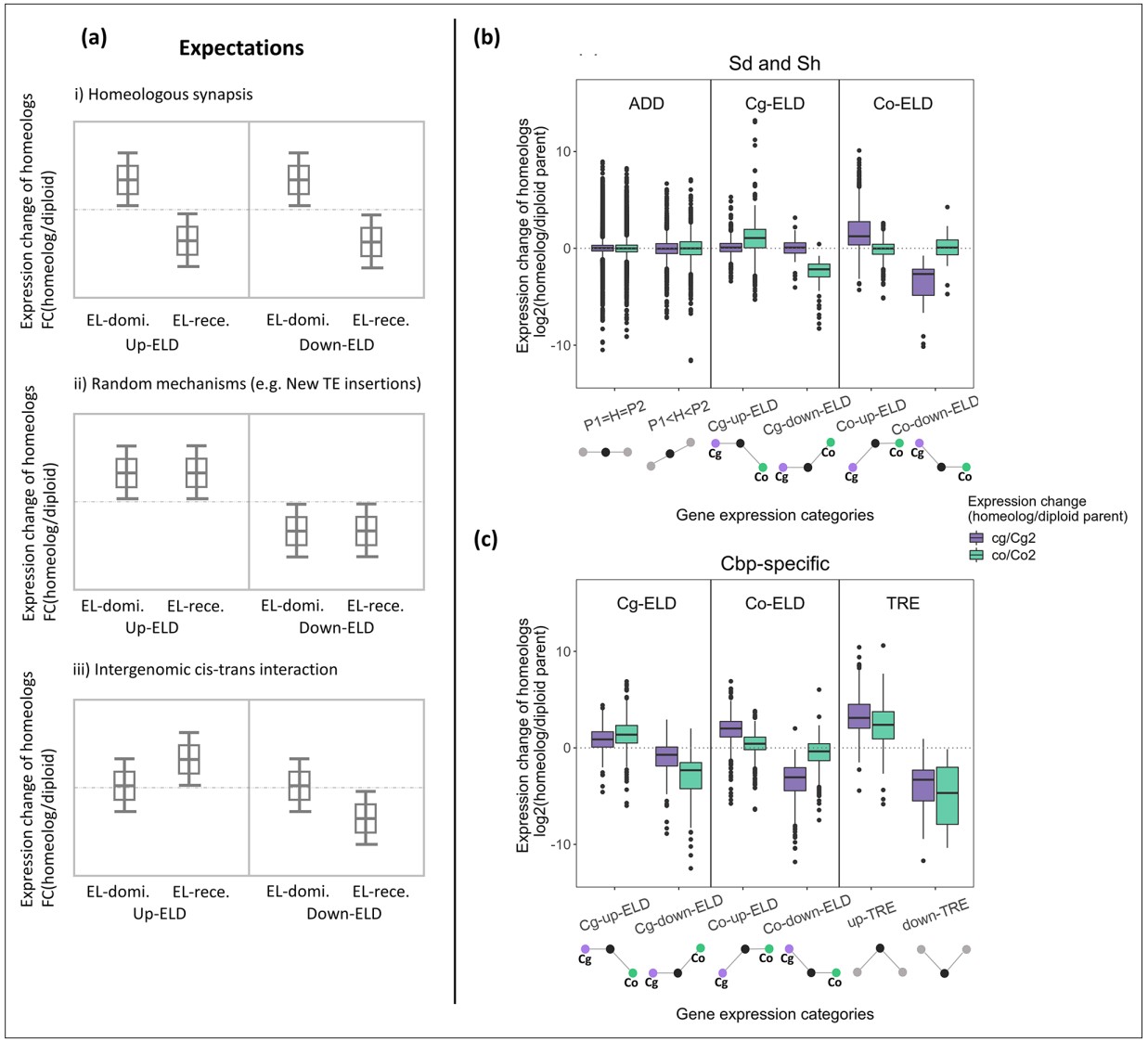

**Figure 8.** Relationships between homoeolog expression change and non-additive gene expression in flowers. (**a**) Expected patterns of homoeolog expression change in each scenario that may explain the cause of expression level dominance (ELD) in resynthesized allotetraploids. (**b**) Observed homoeolog expression change among genes with ELD in resynthesized allotetraploids (Sd and Sh). (**c**) Observed homoeolog expression change among genes with Cbp-specific non-additive expression.

The online version of this article includes the following source data and figure supplement(s) for figure 8:

**Source data 1.** Expression level fold-change (log2FC) of homoeologs relative to the corresponding gene in diploid groups among genes with expression level dominance (ELD) in flowers or leaves.

**Figure supplement 1.** Relationships between homoeolog expression change and non-additive gene expression in leaves.

the direction of ELD, especially in Cg-ELD genes. For genes with Cbp-specific TRE, both homoeologs had expression change in the same direction of TRE (*Figure 8*, *Figure 8—figure supplement 1*, and *Figure 8—source data 1*).

## Discussion

Distinguishing parental legacy from the effects of evolutionary forces is essential for interpreting the outcome of allopolyploidization. The short-term genomic interactions in allopolyploids reflect the divergence of parental genomes (*Johnson, 2010*). In this sense, short-term transcriptomic changes in new allopolyploids are also part of parental legacy but are not observable with only information from

diploid parental species. In this study, we used resynthesized *Capsella* allotetraploids as an approximation of the early stage of the natural allotetraploid species to separate and compare the short- and long-term transcriptomic and phenotypic changes. The timing and pattern of the variation also provided hints for locating the exact mechanism.

The extant diploid species were not the exact parental populations of natural *C. bursa-pastoris*, and the sampled diploid individuals were genetically closer to the resynthesized allotetraploids than to natural *C. bursa-pastoris*, potentially leading to an overestimation of the contribution of long-term mechanisms. However, the divergence between *C. grandiflora* and *C. orientalis* was much more ancient than the formation of *C. bursa-pastoris* (*Douglas et al., 2015*). Therefore, the molecular divergence between *C. grandiflora* and *C. orientalis* after the formation of *C. bursa-pastoris* is only a small fraction of the total divergence.

Besides, the mating system of the real parental populations of *C. bursa-pastoris* was likely the same as today: A nonfunctional self-incompatibility haplotype that was fixed in *C. orientalis* was also found in *C. bursa-pastoris*, suggesting that *C.orientalis* became self-compatible before the formation of *C. bursa-pastoris* (*Bachmann et al., 2019*); On the other hand, restoring the great polymorphism of functional self-incompatibility haplotypes from a self-compatible ancestral population is very unlikely, therefore self-incompatibility should be the ancestral state of the (*C. grandiflora* +*C. rubella*) lineage. Although we cannot exclude the alternative hypothesis that the progenitor of Cbp_cg is a self-fertilizing ghost population, the most parsimonious hypothesis is that Cbp_cg subgenome originated from outcrossing individuals. For all these reasons, the resynthesized *Capsella* allotetraploids may still provide a realistic approximation to the early stages of natural *C. bursa-pastoris*.

Another limitation of using resynthesized allotetraploids is that we could not completely exclude the effect of colchicine treatment, even though we used second-generation allotetraploids (*Münzbergová, 2017*). Colchicine treatment could affect pollen and seed quality and the rate of homoeologous synapsis in resynthesized allotetraploids. Spontaneous *Capsella* allotetraploids have been repeatedly found among diploid hybrids that were not treated with colchicine solution (*Bachmann et al., 2021* own unpublished results). For future studies, these spontaneous allotetraploids would be excellent materials for accurately estimating the rate of homoeologous synapsis in newly formed *Capsella* allotetraploids. Nevertheless, the reported influence of colchicine treatments on the second generation of synthetic polyploids was either trivial (*Husband et al., 2016*) or not in the same direction as our results (*Münzbergová, 2017*). Hence, the observed pollen and seed quality reduction and rampant homoeologous synapsis were unlikely pure artifacts from colchicine treatment.

## Resynthesized and natural *Capsella* allotetraploids had distinct phenotypes

The most noticeable morphological difference between resynthesized and natural *Capsella* allotetraploids was related to the selfing syndrome. Compared to second-generation resynthesized allotetraploids, natural *C. bursa-pastoris* had smaller floral organs, more pollen per flower, and a shorter stem length (*Figure 1*). In particular, trait values of petal size and stamen length of the resynthesized allotetraploids had almost no overlap with natural *C. bursa-pastoris* but were similar to the outcrossing progenitor *C. grandiflora*. The shorter stem length in natural *C. bursa-pastoris* may reflect a shorter lifespan, which is also a feature of self-fertilizing species (*Duminil et al., 2009*; *Lesaffre and Billiard, 2020*). If natural *C. bursa-pastoris* indeed originated from the hybridization between *C. grandiflora*-like outcrossing plants and *C. orientalis*-like self-fertilizing plants, the selfing syndrome in *C. bursa-pastoris* does not reflect the instant dominance effect of the *C. orientalis* alleles, but evolved afterward. A remaining question is whether the genetic basis of the selfing syndrome in *C. bursa-pastoris* is the same as in *C. orientalis*. Did the pre-existing selfing syndrome-related alleles from *C. orientalis* facilitate the evolution of selfing syndrome in *C. bursa-pastoris*? Was the selfing syndrome of *C. bursa-pastoris* established by silencing/replacing the *C. grandiflora* alleles or new regulations on both *C. orientalis* and *C. grandiflora* homoeologs?

Although the selfing syndrome in natural *C. bursa-pastoris* was most likely an adaptation to the change in mating system, these morphological changes may be accelerated by the compensation or adaptation to a polyploid state (*Hollister, 2015*). WGD directly increases the size of various types of cells (*Beaulieu et al., 2008*; *Katagiri et al., 2016*; *Snodgrass et al., 2017*; *Wilson et al., 2021*) and disturbs the efficiency of multiple physiological processes (*Drake et al., 2013*; *Bomblies, 2020*).

Compared to newly resynthesized autopolyploids, natural autopolyploid plants often have smaller cell or organ sizes (*Maherali et al., 2009*; *Münzbergová, 2017*; *Landis et al., 2020*), possibly driven by the demand for optimizing physiological processes or resource allocation (*Roddy et al., 2020*; *Domínguez-Delgado et al., 2021*). In the case of allotetraploid *Capsella*, the selection of selfing-syndrome-related traits and the adaptation to a polyploid state (e.g. decreasing the size of cell or organ for physiological efficiency or better energy allocation) may work synergistically and can be difficult to separate.

Apart from the selfing-syndrome-related traits, newly resynthesized *Capsella* allotetraploids had lower proportions of viable pollen (*Figure 1l*) and normal seeds (*Figure 1m*). In contrast, pollen and seed quality in natural *C. bursa-pastoris* were much higher and as good as in diploid species. The higher pollen and seed quality in natural *C. bursa-pastoris* was possibly achieved by improving meiotic behaviors. Meiotic stabilization is another important aspect of adaptation to an allopolyploid state (*Blasio et al., 2022*). Newly resynthesized allopolyploids suffer more often than natural allopolyploids, from frequent and severe meiotic abnormalities, which are associated with lower pollen viability and fertility in resynthesized allopolyploids (*Szadkowski et al., 2010*; *Zhang et al., 2013*; *Henry et al., 2014*). The molecular basis of improved meiotic synapsis in natural allopolyploids is not completely clear, but several loci that suppress homoeologous synapsis or recombination are essential for the process (*Jenczewski et al., 2003*; *Nicolas et al., 2009*; *Greer et al., 2012*; *Martín et al., 2017*). The exact mechanism of meiotic stabilization in natural *C. bursa-pastoris* needs further investigation.

## The emergence of non-additive gene expression in allotetraploids was a two-stage process

Non-additive gene expression in allotetraploid *Capsella* was altogether limited and neither a complete relict of short-term genomic interactions nor entirely due to gradual divergence. We found that about 40% of the cases of ELD in natural *C. bursa-pastoris* could already be found in the second generation of resynthesized allotetraploids (relict ELDs). Most of these relict ELDs and their directions were shared by the two resynthesized allotetraploid groups (*Figure 4c*, *Figure 4—figure supplement 1*), suggesting that most relict ELDs were repeatable alterations. On the other hand, about 60% of the cases of ELD were specific to natural *C. bursa-pastoris* (Cbp-specific ELDs, *Figure 4c*, and *Figure 4—figure supplement 1*), revealing the contributions from long-term evolution.

The relict ELDs and Cbp-specific ELDs differed in several features. While the vast majority of relict ELDs were up-regulated (97% in flowers and 88% in leaves), Cbp-specific ELDs had a more balanced number of up- and down-regulated ELDs (61% and 54% were up-regulated in flowers and leaves, respectively), suggesting the short and long-term ELD had different molecular basis. In diploid or polyploid interspecific hybrids, overexpression is often more common than underexpression. The trend has been observed in a wide range of organisms, including *Brassica* (*Wu et al., 2018*; *Li et al., 2020*; *Wei et al., 2021*) and *Raphanobrassica* (*Ye et al., 2016*), cotton (*Yoo et al., 2013*), brown algae (*Sousa et al., 2019*), and copepod (*Barreto et al., 2015*). Results in *Capsella* further showed that short-term mechanisms mainly caused the excess of up-regulated ELDs. Among the short-term mechanisms, intergenomic interaction of regulatory elements is the most likely candidate for generating the excess of up-regulated ELDs, considering that these up-regulated ELDs were highly shared between the two resynthesized allotetraploid groups, and between resynthesized and natural allotetraploids. A global DNA methylation change may also contribute to the excess of up-regulation in resynthesized allotetraploids, if methylation levels were systematically lower in *Capsella* allotetraploids, as observed in *Mimulus* (*Edger et al., 2017*). However, methylation change alone fails to explain why the majority of these up-regulated ELDs in resynthesized allotetraploids were retained in natural allotetraploids, especially in leaves (*Figure 4c*, *Figure 4—figure supplement 1*).

Besides, the relict ELDs contained more Cg-ELDs than Co-ELD, but Cbp-specific ELDs had more Co-ELDs, especially in flowers (*Figure 4c*, *Figure 4—figure supplement 1*). The increase of Co-ELDs in natural *C. bursa-pastoris* mirrored the morphological difference: The floral organ size of resynthesized allotetraploids was similar to that of *C. grandiflora*, whereas natural *C. bursa-pastoris* was more similar to *C. orientalis* (*Figure 1a–g*). However, it is worth noting that the reversed trend of ELD may not be the fundamental genetic basis of selfing syndrome, but reflect the different composition of tissue/cells in RNA samples. Both morphological changes and the direction of ELD could result from upstream regulatory changes.

In addition, although unbalanced homoeolog content (not 2:2) caused by homoeologous synapsis was common in our resynthesized allotetraploids, they were still not the main cause of ELD in resynthesized allotetraploids. If ELD were mainly caused by possessing more than two copies of the EL-dominant homoeolog, we would expect the relative expression from the EL-dominant homoeolog to increase and the EL-recessive homoeolog to decrease in genes with significant ELD. In contrast to this expectation, we found that, on average, the expression of EL-dominant homoeologs (relative to the transcriptome of the subgenome) was similar to that in diploid parental species, while the expression of EL-recessive homoeologs changed toward the EL-dominant homoeolog (*Figure 8*). This suggests that ELD is mainly achieved by altering the expression of EL-recessive homoeologs. This result is consistent with studies in a wide range of allopolyploid organisms (*Yoo et al., 2013*; *Cox et al., 2014*; *Combes et al., 2015*; *Sousa et al., 2019*), although not in resynthesized *Brasssica napus* (*Wu et al., 2018*). This conservative pattern can be explained by intergenomic interaction between divergent regulatory elements (*Hu and Wendel, 2019*), but direct evidence is still lacking.

As for transgressive gene expression, we found almost no TRE genes in resynthesized allotetraploids, but a mere 1.3% TRE genes in natural *C. bursa-pastoris*, with a threshold of FC >2 (*Figure 4—source data 1*). In agreement with several previous observations (*Flagel and Wendel, 2010*; *Yoo et al., 2013*; *Zhang et al., 2016b*), the results in *Capsella* suggest that transcriptional novelties in allopolyploids were not an instant outcome of allopolyploidization but mainly arose during long-term evolution and remained altogether rather limited.

## Homoeologous synapsis was common in resynthesized *Capsella* allotetraploids, and may still be a source of variation in natural Cbp

Disomic inheritance in allopolyploid species is not established all at once (*Henry et al., 2014*), and strict disomic inheritance may have never been achieved in some allopolyploid species. In contrast to the disomic inheritance and the low level of homoeolog expression loss in natural *C. bursa-pastoris*, we found abundant traces of homoeologous segregation or recombination in all 12 lines of resynthesized *Capsella* allotetraploids, after only one meiosis. The observation is in line with many recent studies in which abundant homoeologous exchanges were found in allopolyploids (*Lloyd et al., 2018*; *Pelé et al., 2018*). The contrast between resynthesized and natural allotetraploids suggested that preferential synapsis was rapidly improved in natural *C. bursa-pastoris*, and/or a balanced number of homoeologs was strongly preferred by selection, otherwise, we would expect a much higher proportion of homoeolog replacement (having four copies of the same homoeolog) after 100,000 year recurrent self-fertilization with tetrasomic/heterosomic inheritance.

Homoeologous synapsis was the major mechanism for the variation of HEB and homoeolog expression loss in resynthesized *Capsella* allopolyploids (*Figures 5 and 7*). Several models have been proposed to explain HEB and biased genome fractionation in allopolyploids, including different TE contents (*Woodhouse et al., 2014*; *Cheng et al., 2016*; *Wendel et al., 2018*), the epigenetic difference of subgenomes (*Li et al., 2014*), different strength of regulatory elements, as a result of enhancer runaway (*Fyon et al., 2015*; *Bottani et al., 2018*). It was also suggested that the initial HEB may be reinforced in long-term evolution, as the homoeolog with a lower initial expression level is subject to weaker purifying selection, and has a larger chance of degeneration (*Woodhouse et al., 2014*). However, in resynthesized *Capsella* allotetraploids, homoeologous synapsis played an important role in generating HEB variation, possibly overshadowing the influence of other mechanisms. This result does not conflict with the observed association between TE content in parental species and genome dominance (*Woodhouse et al., 2014*). While TE contents may have only a minor effect in directly generating HEB variation, they could still be informative in predicting HEB in established allopolyploids, as the presence of TEs in the flanking regions may affect the fitness effect of HEB variation (*Hollister and Gaut, 2009*). In other words, TEs may not function as a strong mutagenic mechanism of HEB variation, but affect the selection on HEB variation, as a form of genetic load.

For established natural allotetraploids, occasional homoeologous synapsis may still be an important source of genetic variation, even long after the allopolyploidization event. Although an earlier allozymic study (*Hurka et al., 1989*) and an approximate Bayesian computation (ABC) with high throughput sequencing data (*Roux and Pannell, 2015*) suggest that natural *C. bursa-pastoris* exhibits disomic inheritance, neither analysis could reject homoeologous synapsis at a lower rate. A very small proportion of homoeologous synapsis may be negligible for inferring the dominant

mode of inheritance, but in terms of causing homoeologous gene loss and creating genetic variation, homoeologous synapsis can still be more influential than point mutations. Due to the inevitable technical variation of RNA-seq and expression variation across genes, we were unable to confidently resolve smaller blocks of unbalanced homoeolog content. Despite the small sample size and the low resolution of RNA-seq data, we noticed that some small genomic blocks with homoeolog replacement were shared by the individuals of the same population but varied among populations of natural *C. bursa-pastoris* (*Figure 6*, *Figure 6—figure supplements 1 and 2*), suggesting that homoeologous synapsis still contribute to expressional variation in natural *C. bursa-pastoris*. Apart from homoeolog synapsis in a single-origin allopolyploid species, unbalanced content of homoeologs could also arise from secondary introgression from diploid parental species. Detailed demographic modeling would be needed for distinguishing the two scenarios.

## Conclusion

In conclusion, together with *Duan et al., 2023*, the present study shows that both short- and long-term mechanisms contributed to transcriptomic and phenotypic changes in natural allotetraploids. However, the initial gene expression changes were largely reshaped during long-term evolution leading to more pronounced morphological changes. Resource limitations and/or adaptation to self-fertilization also, drive flowers' evolution after polyploidization.

## Materials and methods

### Plant material

The present study used five *Capsella* plant groups (*Figure 1*, *Figure 1—source data 1*), including diploid *C. orientalis* (Co2), diploid *C. grandiflora* (Cg2), two types of resynthesized allotetraploids (Sd and Sh), and natural allotetraploids, *C. bursa-pastoris* (Cbp). RNA-sequencing data and most phenotypic data (except floral morphologic traits) of Co2, Cg2, Sd, and Sh groups were from *Duan et al., 2023*. The Sd and Sh allotetraploids only differed in the order of hybridization and whole genome duplication. Allotetraploids of the Sd group were generated by crossing synthetic autotetraploid *C. orientalis* with autotetraploid *C. grandiflora*, whereas the Sh group was generated by inducing WGD in the first generation of diploid hybrids of *C. orientalis×C. grandiflora*. In all interspecific crosses, *C. orientalis* served as maternal plants, mimicking the formation of the natural allotetraploid species, *C. bursa-pastoris* (*Hurka et al., 2012*). All *C. orientalis* plants used in the experiment are descendants of one wild *C. orientalis* individual, and all the *C. grandiflora* plants are descendants of three individuals from one *C. grandiflora* population (*Figure 1c* and *Figure 1—source data 1*).

To compare the resynthesized allotetraploids with natural allotetraploids, natural *C. bursa-pastoris* was added to the present study. Seeds of wild *C. bursa-pastoris* plants were grown in a growth chamber for one generation. Then the second generation of *C. bursa-pastoris* plants was grown in the same experiment together with the other four plant groups used in *Duan et al., 2023*. All five plant groups were grown in a growth chamber under long-day conditions (16 hr light at 22 °C and 8 hr dark at 20 °C, light intensity = 137 uE·m-2·s-1).

Each of the five plant groups was represented by six 'lines', and each line had six individuals, which were full siblings from either self-fertilization (Co2, Cbp, Sh, and Sd groups) or brother-sister mating (Cg2 group). The six lines of *C. bursa-pastoris* were from six populations (*Figure 1c* and *Figure 1—source data 1*), representing three major genetic clusters of the wild *C. bursa-pastoris* (*Kryvokhyzha et al., 2019b*). Each line originated from an independent allopolyploidization event for the Sh and Sd groups. For Co2 and Cg2 groups, 'line' only referred to the offspring of one parental plant (Co2) or a pair of parental plants (Cg2) in the previous generation.

Plants used in the present study and a previously published work (*Duan et al., 2023*) were different subsets of a single experiment. The entire experiment had eight plant groups, including the five plant groups used in the present study (Groups Cg2, Co2, Sh, Sd, and Cbp; 5 groups x 6 lines x 6 individuals = 180 plants), and other three groups that were only used in *Duan et al., 2023*, Groups F, Co4 and Cg4; 3 groups x 6 lines x 6 individuals = 108 plants. All these plants were grown in 36 trays placed on six shelves inside the same growth chamber. Each tray had exactly one plant from each of the eight groups, and the positions of the eight plants within each tray (A-H) were randomized with random. shuffle() method in Python (*Supplementary file 1*). The position of the 36 trays inside the growth room

(1-36) was also random and the positions of all trays were shuffled once again 28 days after germination (randomized with RAND() and sorting in Microsoft Excel Spreadsheet).

## Phenotyping

Floral morphological traits were measured for all five groups on 165 plants 7–14 days after the first flower opened. The rest 15 plants were not measured due to technical errors. Two fully opened young flowers were dissected for each plant, and the floral organs were scanned with a photo scanner (Epson Perfection V370) at 3200 dpi. Seven floral morphological traits were measured on the digital images with Fiji, an open-source platform for biological-image analysis (*Schindelin et al., 2012*), including sepal width, sepal length, petal width, petal length, pistil width, pistil length, and stamen length. For each plant, two flowers were examined. Three sepals, petals and stamina, and one pistil were measured for each flower.

Stem length (total sample size n=171), flowering time (n=170), pollen traits (n=142), and seed traits (n=151) were measured for the five plant groups. Measurements of the Cg2, Co2, Sh, and Sd groups were from *Duan et al., 2023*, and measurements of the Cbp group were added by the present study, which were obtained in the same way as the other four groups. The length of the longest stem (stem length) was measured on dry plants. The number of days from germination start to the opening of the first flower was recorded as flowering time. The number of pollen grains per flower (pollen counts) was calculated by counting 1/60 (Co2, Sd, Sh, and Cbp groups) or 1/120 (Cg2 group) of the total pollen grains of a flower using a hemocytometer. Pollen viability was measured with an aceto-carmine staining method (*Duan et al., 2023*), by examining at least 300 pollen grains per flower. Pollen counts and pollen viability were measured on two flowers of each individual. Seeds from the 11th to 20th fruits were counted and were used to measure the proportion of normal seeds. In the case of the Cg2 and Cg4 groups, seeds were obtained through hand pollination. The first ten fruits were used when not all of these flowers set fruits. Individuals with less than ten fruits were excluded from the analysis. Seeds that were flat or dark and small were considered abnormal.

General linear models (for non-ratio data) or generalized linear models with quasi-binomial error distribution and a logit link function (for pollen viability and seed quality) were fit to each phenotypic trait. The difference in phenotypic traits among the five plant groups was tested with one-way analyses of variance (ANOVA, trait value ~plant group, type-III tests), using R package 'car' version 3.1–2 (*Fox and Weisberg, 2019*). A two-way ANOVA (trait value ~plant group +tray ID, type-III tests) was also tried to test the positional effect. When plant group had a significant effect on a trait, groups with different means were identified by Tukey's HSD test using R packages 'agricolae' version 1.3–6 (*de Mendiburu and Yaseen, 2020*) and 'multcomp' version 1.4–25 (*Hothorn et al., 2008*).

## RNA-sequencing

RNA-sequencing was conducted for the five plant groups (Co2, Cg2, Sd, Sh, and Cbp). For each line, leaf and young inflorescence (flower) samples from one randomly chosen individual were sequenced, resulting in 60 RNA-sequencing samples (5 groups × 6 lines × 2 tissues). RNA-sequencing data of the Co2, Cg2, Sd, and Sh groups were from *Duan et al., 2023*. Data from the Cbp group was added to the present study, but the Cbp samples were collected and sequenced simultaneously with the other four plant groups in 2019. The 8th and 9th leaves were harvested at the emergence of the 11th leaf, and 2–4 inflorescences with only unopened flower buds were collected 7–14 days after the first flower opened. The collected tissue was frozen in liquid nitrogen and stored at –80 °C.

Total RNA was extracted from leaf and flower samples with a cetyl-trimethyl-ammonium-bromide (CTAB)-based method (*Duan et al., 2023*). DNA contamination was further removed by the RNase-Free DNase Set (QIAGEN). RNA libraries were prepared with Illumina TruSeq Stranded mRNA (poly-A selection) kits and sequenced with pair-end reads of 150 bp on three NovaSeq 6000 S4 lanes, by the SNP&SEQ Technology Platform in Uppsala, Sweden. One sequencing library was generated for each diploid sample, and two libraries were generated for each allotetraploid sample.

## Gene and homoeolog expression

Raw RNA-seq reads were mapped to the reference genome of *Capsella rubella* (*Slotte et al., 2013*) using Stampy v.1.0.32 (*Lunter and Goodson, 2011*). The expected divergence between reference and query sequences was set to 0.02, 0.04, and 0.025 for *C. grandiflora*, *C. orientalis,* and

allotetraploids, respectively. Mapping quality was inspected by Qualimap v. 2.2.1 (*Okonechnikov et al., 2016*). The number of reads mapped to each gene was counted by HTSeq v.0.12.4 (*Anders et al., 2015*), using the mode 'union' (hereinafter referred to as 'unphased gene expression'). The average number of mapped reads was 38.4±2.4 and 70.0±3.5 for the diploid and tetraploid samples, respectively (*Supplementary file 2*). For all analyses on unphased gene expression, the mapped reads were downsampled with a custom Python script (*Duan et al., 2023*), so that all five groups had a similar average number of mapped reads.

The expression level of separate homoeologs in allotetraploids was determined by the program HyLiTE v.2.0.2 (*Duchemin et al., 2015*). Alignment results from the software Stampy v.1.0.32 (*Lunter and Goodson, 2011*) of all five groups and the *C. rubella* reference genome (*Slotte et al., 2013*) were used as the input for HyLiTE. HyLiTE performed SNP calling, classified RNA-seq reads of allotetraploids to parental types according to the identified diagnostic variation between the two diploid parental species and generated a table of homoeolog read counts for allotetraploid individuals (hereinafter referred to as 'phased gene expression').

After partitioning the homoeolog expression, the average library size of allotetraploid subgenomes was similar to the library size of diploid groups (one-way ANOVA, $F_{4,91}=1.28$, p=0.29). To reduce bias between phased and unphased expression datasets, when the homoeolog expression of allotetraploids was compared with gene expression in diploid parental species, the expression level of each gene in diploid individuals was rescaled by the proportion of reads that can be phased for the same gene in allotetraploid individuals.

The overall gene expression pattern of five plant groups in each tissue was visualized by MDS analysis, using the R package 'edgeR' (version 3.28.1; *Robinson et al., 2010*) in the R software environment version 3.6.3 (*R Core Team, 2020*). For both phased and unphased gene expression data, genes with count-per-million (CPM) over one in at least two samples were used for the MDS analysis, and expression levels were normalized with the trimmed mean of M-values (TMM) method.

## Differential expression (DE) analysis

DE analysis was conducted on both unphased and phased gene expression data with the R package 'edgeR' (version 3.28.1; *Robinson et al., 2010*), using TMM normalized gene expression levels. A

**Table 1.** Classification of additive and non-additive gene expression pattern in allotetraploids.

| Group | Description | Classification criteria* |
|---|---|---|
| a | Additive expression with no parental differentiation | $Cg_i = x_{ij} = Co_i$ |
| b | Partial ELD or additive expression with parental differentiation | $(Cg_i < x_{ij} < Co_i)$ or $(Co_i < x_{ij} < Cg_i)$ or $(Cg_i \neq Co_i$ and $x_{ij} = Cg_i$ and $x_{ij} = Co_i)$ |
| c | Up-regulated ELD toward Cg2 | $x_{ij} = Cg_i$ and $x_{ij} > Co_i$ |
| d | Down-regulated ELD toward Cg2 | $x_{ij} = Cg_i$ and $x_{ij} < Co_i$ |
| e | Up-regulated ELD toward Co2 | $x_{ij} = Co_i$ and $x_{ij} > Cg_i$ |
| f | Down-regulated ELD toward Co2 | $x_{ij} = Co_i$ and $x_{ij} < Cg_i$ |
| g | Up-regulated TRE with no parental differentiation | $Cg_i = Co_i$ and $x_{ij} > Cg_i$ and $x_{ij} > Co_i$ |
| h | Up-regulated TRE with parental differentiation | $Cg_i \neq Co_i$ and $x_{ij} > Cg_i$ and $x_{ij} > Co_i$ |
| i | Down-regulated TRE with no parental differentiation | $Cg_i = Co_i$ and $x_{ij} < Cg_i$ and $x_{ij} < Co_i$ |
| j | Down-regulated TRE with parental differentiation | $Cg_i \neq Co_i$ and $x_{ij} < Cg_i$ and $x_{ij} < Co_i$ |

*$Cg_i$: expression level of gene i in the Cg2 group; $Co_i$: expression level of gene i in the Co2 group; $x_{ij}$: expression level of gene i in allotetraploid group j, and j∈(Sd, Sh, Cbp); ELD: expression level dominance; TRE: transgressive expression; The significance of differential expression between groups were determined by results of differential expression analysis on unphased gene expression, with a threshold of fold-change >2 and false discovery rate <0.05.

negative binomial generalized linear model (GLM) was fitted to each dataset. Pairwise group contrasts were then made for each GLM model, and gene-wise quasi-likelihood F-tests were conducted to detect expression changes in each contrast. For unphased data, pairwise contrasts were made among the original five groups (Co2, Cg2, Sd, Sh, and Cbp). For phased data, allopolyploid subgenomes were treated as separate groups (Sd_co, Sd_cg, Sh_co, Sh_cg, Cbp_co, and Cbp_cg) and were compared with the two diploid groups (Co2, Cg2). Genes with an expression fold-change (FC) larger than two and a false discovery rate (FDR) less than 0.05 were considered significant DEGs.

## Expression level dominance and transgressive expression

To measure the extent of non-additive expression in allotetraploids, genes were classified into ten expression categories (*Table 1*), by comparing the total expression of both homoeologs in an allo-tetraploid group to the gene expression level in a diploid group. The 10 categories were modified from the classification by *Zhang et al., 2016a*. Results of DE analysis on unphased gene expression (FC >2 and FDR <0.05) were used for the classification.

## Homoeolog expression bias

HEB of gene $i$ in allotetraploid individual $j$ was measured by the proportion of cg-homoeolog expression ($cg_{ij}$) in the total expression of both homoeologs ($cg_{ij}/(cg_{ij} + co_{ij})$). The distribution of HEB was then viewed by individuals, chromosomes, or along genomic coordinates. Signs of the segregation or recombination of homoeologous chromosomes were revealed in resynthesized allopolyploids by the distribution of HEB along genomic coordinates.

The copy number of cg- and co-homoeologs of each gene and the breakpoints between chromosomal segments resulting from homoeologous recombination were estimated with a five-state HMM of gene-wise HEB, using a modified version of the R package 'HMMcopy' version 1.40.0 (*Lai et al., 2022*). The five states corresponded to (0, 1, 2, 3, 4) gene copies from Cg and (4, 3, 2, 1, 0) copies from Co, respectively. The expected optimal values of median HEB ($m$) were set to 0.01, 0.25, 0.5, 0.75, and 0.99 for the five states. The length of segments was controlled by arguments $e$ and *strength*. $e$ is the initial value of the transition probability that the state (copy number of cg homoeolog) does not change between two adjacent genes, and *strength* is the strength of this initial $e$. Smaller values of *strength* increase the flexibility of transition probability, and an extremely large value leads to almost fixed transition probabilities. For natural *C. bursa-pastoris*, $e$ and *strength* were set to (1 − 1e-7) and 1e+7, respectively. For resynthesized allotetraploids, $e$ was increased to (1 − 1e-10), and *strength* was increased to 1e+12, as homoeologous exchanges were expected to be rare for resynthesized allotetraploids which had only experienced one round of meiosis.

The effect of homoeologous segregation and recombination on HEB in resynthesized allotetraploids was analyzed by a generalized linear model with quasi-binomial error distribution and logit link function. Gene-wise HEB was reshaped as binomial data (read counts from cg homoeolog and total counts from both homoeologs). The copy number of cg homoeologs (0, 1, 2, 3, and 4) estimated by HMM segmentation was used as the explanatory variable. The variance of HEB explained by the estimated copy number of cg homoeolog was assessed by the coefficient of determination ($R^2$), calculated with R package 'rsq' version 2.5 (*Zhang, 2022*).

The relationship between non-additive gene expression and homoeolog expression was explored by comparing the estimated expression fold change of each homoeolog among additive and different non-additive expression categories. The expression fold change of homoeologs was measured by homoeolog expression of gene $i$ in allotetraploid group $k$ divided by expression of gene $i$ in the corresponding diploid group, that is $\log2FC(cg_{ik}/Cg2_i)$ or $\log2FC(co_{ik}/Co2_i)$, using the estimated FC from DE analysis on phased data. The difference of expression fold change between EL-dominant and EL-recessive homoeologs was tested by Welch's two-sample t-tests.

## Loss of homoeolog expression

The most extreme HEB occurs when one homoeolog is silenced or lost in allotetraploids. To explore the timing and mechanism of the loss of homoeolog expression in *Capsella*, the number of genes with homoeolog expression loss was counted for each resynthesized or natural allotetraploid. Lowly or occasionally expressed genes were excluded from the analysis to reduce noise from sequencing and phasing. Specifically, if one homoeologous gene had obvious expression (CPM >5) in all six individuals

of the corresponding diploid species, but had almost no expression in one allotetraploid individual (CPM <0.5), the case was considered a loss of homoeologous expression.

## Gene ontology (GO) enrichment analysis

GO enrichment analysis was performed with R package 'topGO' version 2.52.0 (*Alexa and Maintainer, 2023*), using results of DE analysis on unphased gene expression and GO annotations of *C. rubella* downloaded from PlantRegMap (*Tian et al., 2020*). Genes that were differentially expressed in both Cbp-Sd and Cbp-Sh comparisons (FC >1.5 and FDR < 0.05; 578 genes in flowers and 345 in leaves) were used as the test set, and all genes with GO annotation and CPM > 1 in at least two samples were used as the background set (17,718 genes in flowers and 15,631 genes in leaves). GO terms were scored with the 'classic' algorithm and Fisher's exact tests, and p-values were adjusted with the Benjamini-Hochberg procedure in R. GO terms with less than 10 annotated genes were excluded from the analysis.

## Acknowledgements

Computation and data handling were provided by the Swedish National Infrastructure for Computing (SNIC) at Uppmax. We would like to thank Barbara Mable for her critical comments on the manuscript, and Elsa Sundkvist and Mattias Vass for their help with seed collecting and counting. Swedish Research Council (grant 2019–00806 and 2018–05973) Martin Lascoux, Nilsson-Ehle research grant from the Royal Physiographic Society in Lund Tianlin Duan, Erik Philip-Sörensen Foundation Martin Lascoux, Sven and Lily Lawski Foundation Tianlin Duan.

## Additional information

### Funding

| Funder | Grant reference number | Author |
| --- | --- | --- |
| Swedish Research Council | grant 2019-00806 | Martin Lascoux |
| Erik Philip-Sorensen Foundation | | Martin Lascoux |
| Royal Physiographic Society in Lund | Nilsson-Ehle research grant | Tianlin Duan |
| Sven och Lily Lawskis fond | | Tianlin Duan |
| Swedish Research Council | grant 2018-05973 | Martin Lascoux |

The funders had no role in study design, data collection and interpretation, or the decision to submit the work for publication.

### Author contributions

Tianlin Duan, Conceptualization, Resources, Data curation, Software, Formal analysis, Funding acquisition, Validation, Investigation, Visualization, Methodology, Writing - original draft, Writing - review and editing; Adrien Sicard, Conceptualization, Resources, Investigation, Methodology, Writing - review and editing; Sylvain Glémin, Conceptualization, Methodology, Writing - review and editing; Martin Lascoux, Conceptualization, Resources, Supervision, Funding acquisition, Methodology, Project administration, Writing - review and editing

### Author ORCIDs

Tianlin Duan http://orcid.org/0000-0002-8719-7998
Sylvain Glémin http://orcid.org/0000-0001-7260-4573
Martin Lascoux http://orcid.org/0000-0003-1699-9042

Reviewer #1 (Public Review): https://doi.org/10.7554/eLife.88398.3.sa1
Reviewer #2 (Public Review): https://doi.org/10.7554/eLife.88398.3.sa2
Author Response https://doi.org/10.7554/eLife.88398.3.sa3

## Additional files

### Supplementary files

• Supplementary file 1. Tray ID of each *Capsella* plant and the position of eight plants within each tray.

• Supplementary file 2. RNA-sequencing information.

• Supplementary file 3. Gene ontology (GO) terms that were overrepresented in differentially expressed genes (FC >1.5 and FDR <0.05) between resynthesized and natural allopolyploids.

• MDAR checklist

### Data availability

The RNA-sequencing data of natural C. bursa-pastoris generated by this article are available in NCBI Sequence Read Archive (SRA) and can be accessed with BioProject number PRJNA848625. BioSample accessions are listed in Supplementary file 2. Unphased and phased expression datasets, phenotypic measurements, and R scripts used for statistical tests and visualization are available on GitHub (https://github.com/disporum/RNA104_paper2, copy archived at *Disporum, 2023*).

The following dataset was generated:

| Author(s) | Year | Dataset title | Dataset URL | Database and Identifier |
| --- | --- | --- | --- | --- |
| Duan T, Sicard A, Glémin S, Lascoux M | 2022 | Resynthesized Capsella allopolyploids | https://www.ncbi.nlm. nih.gov/bioproject/? term=PRJNA848625 | NCBI BioProject, PRJNA848625 |

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
