## [Editor Report · eLife assessment]

This **important** study offers new insight into how floral and reproductive phenotypes and gene expression evolve in allopolyploids. The authors marshal **compelling** evidence, using well-constructed genetic lines, RNA sequencing, and phenotypic analyses to distinguish the roles of hybridization, whole genome duplication, and subsequent evolution in phenotypes associated with the selfing syndrome and in gene expression. The work will be of interest to researchers working in plant speciation and genomics, as well as those more broadly interested in the effects of genome copy number on phenotypic and expression evolution.

---

## [Referee Report · Reviewer #1 (Public Review)]

This study aims to determine the relative importance of the immediate effects of allopolyploidization from subsequent evolution in phenotypic traits associated with the selfing syndrome and in gene expression traits in the selfing allopolyploid Capsella bursa-pastoris and its diploid progenitors Capsella grandiflora, which is outcrossing, and Capsella orientalis, which is selfing. To do this, they compared five categories of plant: the two progenitors of the allopolyploid, hybrids resynthesized from the progenitors with a whole-genome duplication either before or after the hybridization event, and the naturally occurring allopolyploid.

Two lines of evidence were used: phenotypic data from the plants grown in a common environment, and RNAseq data from a subset of the plants.

The phenotypic data indicate that the selfing syndrome of C. bursa-pastoris likely evolved after the initial allopolyploidization event, and that pollen and seed viability recovered following the allopolyploidization event. The results are compelling but would benefit from small clarifications to the methods and statistics to account for possible positional effects in the growth chamber. Using a linear mixed model rather than a simple ANOVA would solve this problem.

The RNAseq data are used to explore overall expression patterns (using multi-dimensional scaling), patterns of differential expression (additive, dominant, or transgressive), and homeolog expression bias, and to determine the relative contributions of the original allopolyploidization event and subsequent evolution. Statistical cutoffs were used to categorize gene expression patterns, but the description and categorization of these patterns appears to have been largely qualitative, and might be strengthened by including more statistical detail in questions like whether homeologous expression bias did indeed show more variation in resynthesized and evolved allopolyploids.

The study includes evidence that homeolog expression bias (overrepresentation of an allele from one species) results in part from homeologous synapsis (uneven inheritance of chromosome segments). These deviations from patterns consistent with 2:2 inheritance of genomic regions are highly variable between individuals in resynthesized allopolyploids but appeared to be mostly consistent within (but not between) populations in natural C. bursa-pastoris. This is intriguing evidence that segregation can be an important source of variation in allopolyploids. However, it was limited by the difficulty of inferring homeologous recombination breakpoints with RNAseq data because of the scale of recombination in wild populations (rather than resynthesized allopolyploids). In future identifying such breakpoints will be an interesting direction for this and other allopolyploid systems.

This research suggests many follow-up questions. In particular, it may be possible to identify evidence about the mechanism of the original hybridization event. How frequently do unreduced gametes occur in these species, and is it likely that C. bursa-pastoris evolved via a triploid bridge? Exploring the viability, fertility, and phenotypes of triploids produced in both directions could be a valuable future direction.

Future research, or the current study, could also valuably explore what kinds of genes experienced what forms of expression evolution. A brief description of GO terms frequently represented in genes which showed strong patterns of expression evolution might be suggestive of which selective pressures led to the changes in expression in the C. bursa-pastoris lineage, and to what extent they related to adaptation to polyploidization (e.g. cell-cycle regulators), compensating for the initial pollen and seed inviability or adapting to selfing (endosperm- or pollen-specific genes), or adaptation to abiotic conditions.

Overall, this is an interesting and valuable contribution to the field's understanding of how expression evolves in interaction with hybridization and polyploidy. Particularly in combination with the team's previous study on these lines, this experimental design is effective for separating the contributions of hybridization, WGD, and evolution over time.

Update: the authors have thoughtfully and thoroughly updated the manuscript to address all the questions I raised. I appreciate the chance to review this valuable contribution to the scientific literature.

---

## [Referee Report · Reviewer #2 (Public Review)]

The flowering plant Capsella bursa-pastoris is an allotetraploid formed from the genomes of Capsella orientalis and Capsella grandiflora. An outstanding question in the evolution of allotetraploids is the relative contribution of immediate consequences of allopolyploidization vs. long-term evolution after the event. The authors address this question by re-synthesizing the allotetraploid in the lab using the two progenitor species, and comparing its phenotypic and gene expression variation to naturally occurring C. bursa-pastoris. They find evidence primarily for long-term phenotypic evolution towards a selfing syndrome in C. bursa-pastoris, and a combination of short and long-term changes to gene expression.

The manuscript is thorough and provides lots of new insights into the mechanisms driving evolution in allopolyploids. I especially appreciated the detailed examination of different mechanisms driving gene expression variation. There are some important limitations of the experimental design related to independent evolution of the progenitor species and effects of the colchicine treatment used to induce polyploidy, but these are well-addressed in the Discussion.

---

## [Author Response]

The following is the authors’ response to the original reviews.

**Public Reviews:**
The study could also valuably explore what kinds of genes experienced what forms of expression evolution. A brief description of GO terms frequently represented in genes which showed strong patterns of expression evolution might be suggestive of which selective pressures led to the changes in expression in the C. bursa-pastoris lineage, and to what extent they related to adaptation to polyploidization (e.g. cell-cycle regulators), compensating for the initial pollen and seed inviability or adapting to selfing (endosperm- or pollen-specific genes), or adaptation to abiotic conditions. ”

We did not include a gene ontology (GO) analysis in the first place as we did not have a clear expectation on the GO terms that would be enriched in the genes that are differentially expressed between resynthesized and natural allotetraploids. Even if we only consider adaptive changes, the modifications could occur in various aspects, such as stabilizing meiosis, adapting to the new cell size, reducing hybrid incompatibility and adapting to self-fertilization. And each of these modifications involves numerous biological processes and molecular functions. As we could make post-hoc stories for too many GO terms, extrapolating at this stage have limited implications and could be misleading.

Nonetheless, we are not the only study that compared newly resynthesized and established allopolyploids. GO terms that were repeatedly revealed by this type of exploratory analysis may give a hint for future studies. For this reason, now we have reported the results of a simple GO analysis.

Recommendations for the authors: please note that you control which, if any, revisions, to undertakeThe majority of concerns from reviewers and the reviewing editor are in regards to the presentation of the manuscript; that the framing of the manuscript does not help the general reader understand how this work advances our knowledge of allopolyploid evolution in the broad sense. The manuscript may be challenging to read for those who aren't familiar with the study system or the genetic basis of polyploidy/gene expression regulation. Further, it is difficult to understand from the introduction how this work is novel compared to the recently published work from Duan et al and compared to other systems. Because eLife is a journal that caters to a broad readership, re-writing the introduction to bring home the novelty for the reader will be key.Additionally, the writing is quite technical and contains many short-hands and acronyms that can be difficult to keep straight. Revising the full text for clarity (and additionally not using acronyms) would help highlight the findings for a larger audience.
**Reviewer #1 (Recommendations For The Authors):**
Most of my suggestions on this interesting and well-written study are minor changes to clarify the writing and the statistical approaches.The use of abbreviations throughout for both transcriptional phenomena and lines is logical because of word limits, but for me as a reader, it really added to the cognitive burden. Even though writing out "homoeolog expression bias" or "hybridization-first" every time would add length, I would find it easier to follow and suspect others would too.

Thank you for this suggestion. Indeed, using less uncommon acronyms or short-hands should increase the readability of the text for broader audience. Now in most places, we refer to “Sd/Sh” and “Cbp” as “resynthesized allotetraploids” and “natural allotetraploids”, respectively. We have also replaced the most occurrences of the acronyms for transcriptional phenomena (ELD, HEB and TRE) with full phrases, unless there are extra attributes before them (such as “Cg-/Co-ELD” and “relic/Cbp-specific ELD”).

It would be helpful to include complete sample sizes to either a slightly modified Figure 1 or the beginning of the methods, just to reduce mental arithmetic ("Each of the five groups was represented by six "lines", and each line had six individuals" so there were 180 total plants, of which 167 were phenotyped - presumably the other 13 died? - and 30 were sequenced).

The number 167 only applied to floral morphorlogical traits (“Floral morphological traits were measured for all five groups on 167 plants…”), but the exact total sample size for other traits differed. Now the total sample sizes of other traits have also been added to beginning of the second paragraph of the methods.

For this study 180 seedings have been transplanted from Petri dishes to soil, but 8 seedlings died right after transplanting, seemingly caused by mechanical damage and insufficient moistening. Later phenotyping (2020.02-2020.05) was also disrupted by the COVID-19 pandemic, and some individuals were not measured as we missed the right life stages. Specifically, 5 individuals were missing for floral morphological traits (sepal width, sepal length, petal width, petal length, pistil width, pistil length, and stamen length), 30 for pollen traits, 1 for stem length, and 2 for flowering time. As for seed traits, we only measured individuals with more than ten fruits, so apart from the reasons mentioned above, individuals that were self-incompatible and had insufficient hand-pollination were also excluded. We spotted another mistake during the revision: two individuals with floral morphological measurements had no positional information (tray ID). These measurements were likely mis-sampled or mislabeled, and were therefore excluded from analysis. We assumed most of these missing values resulted from random technical mistakes and were not directly related to the measured traits.

In general, the methods did a thorough job of describing the genomics approaches but could have used more detail for the plant growth (were plants randomized in the growth chamber, can you rule out block/position effects) and basic statistics (what statistical software was used to perform which tests comparing groups in each section, after the categories were identified).When describing the methods, mention whether the plants; this should be straightforward as a linear model with position as a covariate.

Data used in the present study and a previously published work (Duan et al., 2023) were different subsets of a single experiment. For this reason, we spent fewer words in describing shared methods in this manuscript but tried to summarize some methods that were essential for understanding the current paper. But as you have pointed out, we did miss many important details that should have been kept. Now we have added some description and a table (Supplementary file 1) in the “Plant material” section for explaining randomization, and added more information of the software used for performing statistic tests in the “Phenotyping” section.

Although we did not mention in the present manuscript, we used a randomized block design for the experiment (Author response image 1).

**Author response image 1.**

**Author response image 1. sa2fig1:** Plant positions inside the growth chamber.

Plants used in the present study and Duan et al. (2023) were different subsets of a single experiment. The entire experiment had eight plant groups, including the five plant groups used in the present study diploid C. orientalis (Co2), diploid C. grandiflora (Cg2), “whole-genome-duplication-first” (Sd) and “hybridization-first”(Sh) resynthesized allotetraploids, and natural allotetraploids, C. bursa pastoris (Cbp), as well as three plant groups that were only used in Duan et al. (2023; tetraploid C. orientalis (Co4), tetraploid C. grandiflora (Cg4) and diploid hybrids (F)). Each of the eight plant groups had six lines and each line represented by six plants, resulting in 288 plants (8 groups x 6 lines x 6 individuals = 288 plants). The 288 plants were grown in 36 trays placed on six shelves inside the same growth chamber. Each tray had exactly one plant from each of the eight groups, and the position of the eight plants within each tray (A-H) were randomized with random.shuffle() method in Python (Supplementary file 1). The position of the 36 trays inside the growth room (1-36) was also random and the positions of all trays were shuffled once again 28 days after germination (randomized with RAND() and sorting in Microsoft Excel Spreadsheet). (a) Plant distribution; (b) An example of one tray; (c) A view inside the growth chamber, showing the six benches.

With the randomized block design and one round of shuffling, positional effect is very unlikely to bias the comparison among the five plant groups. The main risk of not adding positions to the statistical model is increasing error variance and decreasing the statistical power for detecting group effect. As we had already observed significant among-group variation in all phenotypic traits (p-value <2.2e-16 for group effect in most tests), further increasing statistical power is not our primary concern. In addition, during the experiment we did not notice obvious difference in plant growth related to positions. Although we could have added more variables to account for potential positional effects (tray ID, shelf ID, positions in a tray etc.), adding variables with little effect may reduce statistical power due to the loss of degree of freedom.

Due to one round of random shuffling, positions cannot be easily added as a single continuous variable. Now we have redone all the statistical tests on phenotypic traits and included tray ID as a categorical factor (Figure 2-Source Data 1). In general, the results were similar to the models without tray ID. The F-values of group effect was only slightly changed, and p-values were almost unchanged in most cases (still < 2.2e-16). The tray effect (df=35) was not significant in most tests and was only significant in petal length (p-value=0.0111), sepal length (p-value=0.0242) and the number of seeds in ten fruits (p-value=0.0367). As expected, positions (tray ID) had limited effect on phenotypic traits.

Figure 2 - I assume the numbers at the top indicate sample sizes but perhaps add this to the figure caption.

Statistical power depends on both the total sample size and the sample size of each group, especially the group with the fewest observations. We lost different number of measurements in each phenotypic trait, and for pollen traits we did have a notable loss, so we chose to show sample sizes above each group to increase transparency. Since we had five different sets of sample sizes (for floral morphological traits, stem length, days to flowering, pollen traits and seed traits, respectively), it would be cumbersome to introduce all 25 numbers in figure caption and could be hard for readers to match the sample sizes with results. For this reason, we would like to keep the sample sizes in the figure, and now we have modified the legend to clarify that the numbers above groups are sample sizes.

’The trend has been observed in a wide range of organisms, including ...’ - perhaps group Brassica and Raphanobrassica into one clause in the sentence, since separating them out undermines the diversity somewhat.

Indeed, it is very strange to put “cotton” between two representatives from Brassicaceae. Now the sentence is changed to “… including Brassica (Wu et al., 2018; Li et al., 2020; Wei et al., 2021) and Raphanobrassica (Ye et al., 2016), cotton (Yoo et al., 2013)…”

The diagrams under the graph in Figure 4B are particularly helpful for understanding the expression patterns under consideration! I appreciated them a lot!

Thank you for the comment. We also feel the direction of expression level dominance is convoluted and hard to remember, so we adopted the convention of showing the directions with diagrams.

**Reviewer #2 (Recommendations For The Authors):**
The science is very interesting and thorough, so my comments are mostly meant to improve the clarity of the manuscript text:I found it challenging to remember the acronyms for the different gene expression phenomena and had to consistently cross-reference different parts of the manuscript to remind myself. I think using the full phrase once or twice at the start of a paragraph to remind readers what the acronym stands for could improve readability.

Thank you for this reasonable suggestion. Now we have replaced the most occurrence of acronyms with the full phrases.

There are some technical terms, such as "homoeologous synapsis" and "disomic inheritance", which I think are under-defined in the current text.

Indeed these terms were not well-defined before using in the manuscript. Now we have added a brief explanation for each term.

Under the joint action of these forces, allopolyploid subgenomes are further coordinated and degenerated, and subgenomes are often biasedly fractionated" This sentence has some unclear terminology. Does "coordinated" mean co-adapted, co-inherited, or something else? Is "biasedly fractionated" referring to biased inheritance or evolution of one of the parental subgenomes?

We apologize for not using accurate terms. With “coordinated” we emphasized the evolution of both homoeologs depends on the selection on total expression of both homoeologs, and on both relative and absolute dosages, which may have shifted away from optima after allopolyploidization. “Co-evolved” or “co-adapted” might be a better word.

But the term "biasedly fractionation" has been commonly used for referring to the phenomenon that genes from one subgenome of polyploids are preferentially retained during diploidization (Woodhouse et al., 2014; Wendel, 2015). Instead of inventing a new term, we prefer to keep the same term for consistency, so readers could link our findings with numerous studies in this field. Now the sentence is changed to “Under the joint action of these forces, allopolyploid subgenomes are further co-adapted and degenerated, and subgenomes are often biasedly retained, termed biased fractionation”.

There are a series of paragraphs in the results, starting with "Resynthesized allotetraploids and the natural Cbp had distinct floral morphologies", which consistently reference Figure 1 where they should be referencing Figure 2.

Thank you for spotting this mistake! Now the numbers have been corrected.

‘The number of pollen grains per flower decreased in natural Cbp’ this wording implies it's the effect of some experimental treatment on Cbp, rather than just measured natural variation.

Yes, it is not scientifically precise to say this in the Results section, especially when describing details of results. We meant that assuming resynthesized allopolyploids are good approximation of the initial state of natural allotetraploid C. bursa-pastoris, our results indicate that the number of pollen grains had decreased in natural C. bursa-pastoris. But this is an implication, rather than an observation, so the sentence is better rewritten as “Natural allotetraploids had less pollen grains per flower.”

‘The percentage of genes showing complete ELD was altogether limited but doubled between resynthesized allotetraploid groups and natural allotetraploids’ for clarity, I would suggest revising this to something like "doubled in natural allotetraploids relative to resynthesized allotetraploids

Thank you for the suggestion. The sentence has been revised as suggested.

I'm not sure I understand what the difference is between expression-level dominance and homeolog expression bias. It seems to me like the former falls under the umbrella of the latter.

Expression-level dominance and homeolog expression bias are easily confused, but they are conceptually independent. One gene could have expression-level dominance without any homeolog expression bias, or strong homeolog expression bias without any expression-level dominance. The concepts were well explained in Grover et al., (2012) with nice figures.

Expression level dominance compares the total expression level of both homoeologs in allopolyploids with the expression of the same gene in parental species, and judges whether the total expression level in allopolyploids is only similar to one of the parental species. The contributions from different homoeologs are not distinguished.

While homoeolog expression bias compares the relative expression level of each homoeologs in allopolyploids, with no implication on the total expression of both homoeologs.

Let the expression level of one gene in parental species X and Y be e(X) and e(Y), respectively. And let the expression level of x homoeolog (from species X) and y homoeolog (from species Y) in allopolyploids be e(x) and e(y), respectively.

Then a (complete) expression level dominance toward species X means:e(x)+e(y)=e(X) and e(x)+e(y)≠e(Y);

While a homoeolog expression bias toward species X means:e(x) > e(y), or e(x)/e(y) > e(X)/e(Y), depending on the definition of studies.

Both expression-level dominance and homeolog expression bias have been widely studied in allopolyploids (Combes et al., 2013; Li et al., 2014; Yoo et al., 2014; Hu & Wendel, 2019). As the two phenomena could be in opposite directions, and may be caused by different mechanisms, we think adopting the definitions in Grover et al., (2012) and distinguishing the two concepts would facilitate communication.

Is it possible to split up the results in Figure 7 to show which of the two homeologs was lost (i.e. orientalis vs. grandiflora)? Or at least clarify in the legend that these scenarios are pooled together in the figure?

Maybe using acronyms without explanation made the figure titles hard to understand, but in the original Figure 7 the loss of two homoeologs were shown separately. Figure 7a,c showed the loss of C. orientalis-homoeolog (“co-expession loss”), and Figure 7b,d showed the loss of C. grandiflora-homoeolog (“cg-expession loss”). Now the legends have been modified to explain the Figure.

The paragraph starting with "The extant diploid species" is too long, should probably be split into two paragraphs and edited for clarity.

The whole paragraph was used to explain why the resynthesized allotetraploids could be a realistic approximation of the early stage of C. bursa-pastoris with two arguments:

1. The further divergence between C. grandiflora and C. orientalis after the formation of C. bursa-pastoris should be small compared to the total divergence between the two parental species;

2. The mating systems of real parental populations were most likely the same as today.Now the two arguments were separated as two paragraphs, and the second paragraph has been shortened.

On the other hand, the number of seeds per fruit" implies this is evidence for an alternative hypothesis, when I think it's really just more support for the same idea.

“On the other hand” was used to contrast the reduced number of pollen grains and the increased number of seeds in natural allotetraploids. As both changes are typical selfing syndrome, indeed the two support the same idea. We replaced the “On the other hand” with “Moreover”.

‘has become self-compatible before the formation" "has become" should be "became".

The tense of the word has been changed.

If natural C. bursa-pastoris indeed originated from the hybridization between C. grandiflora-like outcrossing plants and C. orientalis-like self-fertilizing plants, the selfing syndrome in C. bursa-pastoris does not reflect the instant dominance effect of the C. orientalis alleles, but evolved afterward.’ This sentence should be closer to the end of the paragraph, after the main morphological results are summarized.

Thank you for the suggestion. The paragraph is indeed more coherent after moving the conclusion sentence.

References

Combes, M.C., Dereeper, A., Severac, D., Bertrand, B. & Lashermes, P. (2013) Contribution of subgenomes to the transcriptome and their intertwined regulation in the allopolyploid Coffea arabica grown at contrasted temperatures. New Phytologist, 200, 251–260.

Grover, C.E., Gallagher, J.P., Szadkowski, E.P., Yoo, M.J., Flagel, L.E. & Wendel, J.F. (2012) Homoeolog expression bias and expression level dominance in allopolyploids. New Phytologist, 196, 966–971.

Hu, G. & Wendel, J.F. (2019) Cis – trans controls and regulatory novelty accompanying allopolyploidization. New Phytologist, 221, 1691–1700.

Li, A., Liu, D., Wu, J., Zhao, X., Hao, M., Geng, S., et al. (2014) mRNA and Small RNA Transcriptomes Reveal Insights into Dynamic Homoeolog Regulation of Allopolyploid Heterosis in

Nascent Hexaploid Wheat. The Plant Cell, 26, 1878–1900.Wendel, J.F. (2015) The wondrous cycles of polyploidy in plants. American Journal of Botany, 102, 1753–1756.

Woodhouse, M.R., Cheng, F., Pires, J.C., Lisch, D., Freeling, M. & Wang, X. (2014) Origin, inheritance, and gene regulatory consequences of genome dominance in polyploids. Proceedings of the National Academy of Sciences of the United States of America, 111, 5283–5288.

Yoo, M.J., Liu, X., Pires, J.C., Soltis, P.S. & Soltis, D.E. (2014) Nonadditive Gene Expression in Polyploids. https://doi.org/10.1146/annurev-genet-120213-092159, 48, 485–517.